# Synthesis, Antifungal Activity, 3D-QSAR, and Molecular Docking Study of Novel Menthol-Derived 1,2,4-Triazole-thioether Compounds

**DOI:** 10.3390/molecules26226948

**Published:** 2021-11-17

**Authors:** Mei Huang, Wen-Gui Duan, Gui-Shan Lin, Bao-Yu Li

**Affiliations:** 1School of Chemistry and Chemical Engineering, Guangxi University, Nanning 530004, China; 1514404003@st.gxu.edu.cn (M.H.); 2114402005@st.gxu.edu.cn (B.-Y.L.); 2Guangxi Research Institute of Chemical Industry Co., Ltd., Nanning 530001, China

**Keywords:** menthol, 1,2,4-triazole-thioether, antifungal activity, 3D-QSAR, molecular docking

## Abstract

A series of novel menthol derivatives containing 1,2,4-triazole-thioether moiety were designed, synthesized, characterized structurally, and evaluated biologically to explore more potent natural product-based antifungal agents. The bioassay results revealed that at 50 μg/mL, some of the target compounds exhibited good inhibitory activity against the tested fungi, especially against *Physalospora piricola*. Compounds **5b** (R = *o*-CH_3_ Ph), **5i** (R = *o*-Cl Ph), **5v** (R = *m*,*p*-OCH_3_ Ph) and **5x** (R = *α*-furyl) had inhibition rates of 93.3%, 79.4%, and 79.4%, respectively, against *P. piricola*, much better than that of the positive control chlorothalonil. Compounds **5v** (R = *m*,*p*-OCH_3_ Ph) and **5g** (R = *o*-Cl Ph) held inhibition rates of 82.4% and 86.5% against *Cercospora arachidicola* and *Gibberella zeae*, respectively, much better than that of the commercial fungicide chlorothalonil. Compound **5b** (R = *o*-CH_3_ Ph) displayed antifungal activity of 90.5% and 83.8%, respectively, against *Colleterichum orbicalare* and *Fusarium oxysporum* f. sp. *cucumerinum*. Compounds **5m** (R = *o*-I Ph) had inhibition rates of 88.6%, 80.0%, and 88.0%, respectively, against *F. oxysporum* f. sp. *cucumerinu*, *Bipolaris maydis* and *C. orbiculare*. Furthermore, compound **5b** (R = *o*-CH_3_ Ph) showed the best and broad-spectrum antifungal activity against all the tested fungi. To design more effective antifungal compounds against *P. piricola*, 3D-QSAR analysis was performed using the CoMFA method, and a reasonable 3D-QSAR model (*r*^2^ = 0.991, *q*^2^ = 0.514) was established. The simulative binding pattern of the target compounds with cytochrome P450 14α-sterol demethylase (CYP51) was investigated by molecular docking.

## 1. Introduction

Fungicides have been widely used for controlling the disease caused by plant pathogens. However, the developed resistance of plant-pathogenic fungi to currently commercial fungicides has shown great threat to the production and quality of plants, which lead to the exploration of novel antifungal agents.

Peppermint (*Mentha Canadensis* Linnaeus) is an important herb spice plant mainly planted in Jiangsu and Anhui regions of China, as well as a traditional Chinese medicine. Menthol, (1*R*, 2*S*, 5*R*)-2- isopropyl-5-methylcyclohexan-1-ol, is the dominant constituent of the essential oil which is obtained by steam distillation of the fresh leaves of peppermint [1]. The content of menthol in peppermint essential oil can be as high as 30% to 55% [2]. Because menthol is a monocyclic monoterpene alcohol with three chiral carbons, it can be employed to synthesize bioactive compounds except its use in flavors and fragrances. Menthol and its derivatives have a wide range of applications in medicine and agrochemical, such as anti-inflammatory [3], antimicrobial [4,5], antifungal [6], insecticidal [7], anticancer [8,9], analgesic [10,11], anesthetic [12] and penetration-enhancing activities [13]. Thus, menthol deserves further study for agrochemical or pharmaceutical use based on its bioactive property and chemical reactivity.

On the other hand, 1,2,4-triazole, as the representative of five-membered unsaturated heterocycles, showed diverse biological activities, such as antimicrobial [14,15], antifungal [16,17,18], anticancer [19,20], insecticidal [21,22], herbicidal [23,24] and anti-inflammatory activities [25]. In particular, several compounds such as cytochrome P450 14α-sterol demethylase (CYP51) inhibitors in the FRAC classification of fungicides (http://www.frac.info/ accessed on 7 November 2021) were found to contain 1,2,4-triazole moiety. In addition, thioether derivatives have considerable applications in agriculture and medicine because of their various bioactive activities such as insecticidal [26], antifungal, antibacterial [27] and anticancer [28] activities. 

In continuation of our interest in the bioactive properties of natural product-based compounds [29,30,31,32,33,34,35,36,37], a series of novel menthol-derived 1,2,4-triazole-thioether compounds were synthesized by integrating bioactive 1,2,4-triazole and thioether moieties into the molecular skeleton of menthol. Structural characterization, antifungal evaluation, three-dimensional quantitative structure-activity relationship (3D-QSAR) and molecular docking of the title compounds were also carried out.

## 2. Results and Discussion

### 2.1. Synthesis and Characterization

As illustrated in Figure 1, (-)-menthyl-2-chloroacetate (**2**) was prepared by the *O*-acylation reaction of (-)-menthol (**1**) with chloroacetyl chloride. Meanwhile, a series of substituted aryl hydrazides were converted to the corresponding 5-substituted 1,2,4-triazole-3-thiones (**4**). Lastly, a series of menthol-derived 1,2,4-triazole-thioether compounds (**5a-****z**) were synthesized by the nucleophilic substitution of compounds **4** with **2** under alkaline condition.

The structures of all the target compounds and the key intermediate **2** were characterized by FT-IR, ^1^H NMR, ^13^C NMR, electrospray mass spectrometry (ESI-MS) and elemental analysis, and the related spectra could be found in the Appendix A. In the FT-IR spectra of the target compounds, the characteristic absorption bands at about 1731 cm^−1^ were assigned to the stretching vibrations of C=O. The characteristic absorption bands at about 1457–1486 cm^−1^ were assigned to C=N in the 1,2,4-triazole moiety. Additionally, the characteristic absorption bands in the region of 686–714 cm^−1^ was attributed to the vibrations of C-S-C. In the ^1^H-NMR spectra, the protons of benzene ring showed signals at 6.57–7.95 ppm. The characteristic signals at 3.38–3.86 ppm were assigned to the methyl protons linked to the 1,2,4-triazole heterocycle. The methylene protons bonded to the S atom displayed the signals at about 3.40 ppm. The characteristic signals at about 4.71 ppm were assigned to the protons on the saturated carbon bonded to the O atom. The protons of the menthol moiety displayed signals in the range of 0.71–2.02 ppm. The ^13^C NMR spectra of the target compounds showed peaks for the carbon atom of C=O at about 168.00 ppm and carbon atoms of the benzene ring at 96.62–163.92 ppm. For the 1,2,4-triazole moiety, the signals at 150.36–157.37 ppm and 144.06–151.45 ppm were assigned to the unsaturated carbons, and at 31.41–33.40 ppm to the methyl. The methylene carbon bonded to the S atom showed signals at 35.77–36.35 ppm. The carbon bonded to the O atom displayed signals at about 76.45 ppm. The other saturated carbons of menthol displayed signals in the region of 16.22–47.06 ppm. Their molecular weights and the C, H, and N elemental ratios agreed with the results of ESI-MS and elemental analysis, respectively.

### 2.2. Antifungal Activity

The antifungal activities of the target compounds menthol-derived 1,2,4-triazole-thioethers **5a**–**5z** were evaluated by the in vitro method against fusarium wilt on cucumber (*Fusarium oxysporum* f. sp. *cucumerinum*), speckle on peanut (*Cercospora arachidicola*), apple ring rot (*Physalospora piricola*), tomato early blight (*Alternaria solani*), wheat scab (*Gibberella zeae*), rice sheath blight (*Rhizoeotnia solani*), corn southern leaf blight (*Bipolaris maydis*), and watermelon anthracnose (*Colleterichum orbicalare*) at 50 µg/mL, using the commercial antifungal agent chlorothalonil as positive control. The results are listed in Table 1.

The bioassay results indicated that at the concentration of 50 µg/mL, the target compounds **5a**–**5z** exhibited certain antifungal activity against the eight tested fungi, especially against *P. piricola*. Compounds **5b** (R = *o*-CH_3_ Ph), **5i** (R = *o*-Cl Ph), **5v** (R = *m*,*p*-OCH_3_ Ph) and **5x** (R = *α*-furyl) had inhibition rates of 93.3%, 79.4%, 79.4% and 79.4%, respectively, against *P. piricola*, better than that of the positive control chlorothalonil with inhibition rate of 75.0%. Compounds **5v** (R = *m*,*p*-OCH_3_ Ph), **5i** (R = *o*-Cl Ph) and **5o** (R = *o*-CF_3_ Ph) held inhibition rates of 82.4%, 79.0% and 79.0% against *C. arachidicola*, respectively, better than that of the commercial fungicide chlorothalonil with inhibition rate of 73.3%. Compound **5g** (R = *m*-F Ph) displayed inhibitory activity of 86.5% against *G. zeae*, better than that of the positive control chlorothalonil with inhibition rate of 73.1%. Compound **5b** (R = *o*-CH_3_ Ph) displayed antifungal activity of 90.5% and 83.8%, respectively, against *C. orbicalare* and *F. oxysporum* f. sp. *cucumerinu*. Compounds **5m** (R = *o*-I Ph) had inhibition rates of 88.6%, 80.0%, and 88.0%, respectively, against *F. oxysporum* f. sp. *cucumerinu*, *B. myadis* and *C. orbiculare*. In addition, compound **5b** (R = *o*-CH_3_ Ph) displayed the best and broad-spectrum antifungal activity against all the tested fungi. Therefore, compound **5b** (R = *o*-CH_3_ Ph) deserved further study. To further investigate the structure–activity relationship, 3D-QSAR and molecular docking studies were subsequently performed. 

### 2.3. CoMFA Analysis

The 3D-QSAR analysis of the antifungal activity against *P. piricola* of the target compounds was carried out by the CoMFA method to study the structure-activity relationship of the synthesized compounds. In this work, 23 derivatives were selected to perform 3D-QSAR study and the results of CoMFA model are listed in Table 2. This model with cross-validated coefficient (*q*^2^ = 0.514 > 0.5), correlation coefficient (*r*^2^ = 0.991 > 0.8), standard error of estimate (*S* = 0.050) and Fischer ratio (*F* value = 183.384) was reliable and effective. In addition, the Scatter plot of predicted ED values vs experimental ED values shown in Figure 1 was drawn by the data presented in Table 3. In Figure 1, it was easily observed that the all points were placed at the nearby of Y = X line.

The electrostatic and steric contribution maps of CoMFA are shown in Figure 2. The contribution rate of the electrostatic field and the steric field were 24.8% and 75.2%, respectively, revealing that the steric field was the major contributor to the increase in the antifungal activity against *P. piricola*. In Figure 2A, there were some green regions located around the 2-position of the benzene ring indicating that the introduction of bulky group in this position was beneficial to increase the antifungal activity. For example, compounds **5b** (R = *o*-CH_3_ Ph), **5d** (R = *o*-OCH_3_ Ph), **5f** (R = *o*-F Ph), **5i** (R = *o*-Cl Ph), **5m** (R = *o*-I Ph), **5o** (R = *o*-CF_3_ Ph), **5q** (R = *o*-OH Ph) and **5s** (R = *o*-NH_2_ Ph) displayed higher antifungal activity than compound **5a** (R = Ph). In contrast, the yellow regions (Figure 2A) at the 4-position of the phenyl ring, indicated that the introduction of bulky group in this position was associated with worse antifungal activities. For example, compounds **5c** (R = *p*-CH_3_ Ph), **5h** (R = *p*-F Ph), **5k** (R = *p*-Cl Ph), **5l** (R = *p*-Br Ph) and **5r** (R = *p*-OH Ph) displayed worse antifungal activity than compound **5a** (R = Ph). In Figure 2B, the electrostatic field contours were displayed in two distinguishing colors. The blue block represents that the introduction of electron donor groups is conducive to enhancing activity and the opposite for the red block. Therefore, the introduction of electron-donating groups at the 4-position and electron-withdrawing groups at the 2-position of benzene rings were both favorable for antifungal activity. For instance, compound **5c** (R = *p*-CH_3_ Ph) and **5e** (R = *p*-OCH_3_ Ph) exhibited better antifungal activity than that of **5h** (R = *p*-F Ph), **5k** (R = *p*-Cl Ph) and **5l** (R = *p*-Br Ph), compounds **5f** (R = *o*-F Ph), **5i** (R = *o*-Cl Ph), and **5m** (R = *o*-I Ph) possessed a higher inhibitory rate than **5d** (R = *o*-OCH_3_ Ph).

### 2.4. Molecular Docking Analysis

For investigating the binding modes of the target compounds in the active site of cytochrome P450 14α-sterol demethylase (CYP51), molecular docking study was carried out using Sybyl-X 2.1.1 software. As a representative sample, the binding modes of compound **5b** (R = *o*-CH_3_ Ph) are respectively presented in the 2D and 3D pattern (Figure 3). It was found that compound **5b** (R = *o*-CH_3_ Ph) could readily embed into the binding pocket and interact with the residues around. In particular, the formed hydrogen bond between the O atom of the ester group of the ligand and the hydroxyl of the residue Tyr76(A) was observed. In addition, the small-molecule ligand also generated hydrophobic interactions with other residues, such as Phe78(A), Met79(A), Leu100(A), and Leu321(A). For comparison, imibenconazole, a commercial CYP51 inhibitor, was also evaluated for its binding modes in the active site, and the obtained results were shown in the 2D pattern (Figure 3D). Most of the residues around compound **5b** (R = o-CH_3_ Ph) could be also observed in the case of imibenconazole, revealing that they exhibited similar binding modes in the active site. 

Moreover, their binding affinity with the target protein could be expressed by total score [38], which was automatically generated from the docking process. To our delight, all of the total scores of the target compounds (6.5160–9.1989) were higher than that of imibenconazole (5.4657), indicating that the target compounds showed stronger binding affinity with CYP51 than imibenconazole.

## 3. Experimental Section

### 3.1. General Information

The GC analysis was conducted on an Agilent 6890 GC (Agilent Technologies Inc., Santa Clara, CA, USA) equipped with column HP-1 (30 m, 0.530 mm, 0.88 μm). The IR spectra were recorded on a Nicolet iS50 FT-IR spectrometer (Thermo Scientific Co., Ltd., Madison, WI, USA) using the KBr pellet method. The HPLC analysis was conducted on Waters 1525 HPLC instrument (Waters Co., Ltd., Milford, MA, USA) equipped with column C18 5 µm (4.6 mm × 150 mm) and Waters 2998 PDA detector. The melting points were recorded using an MP420 automatic melting point apparatus (Hanon Instruments Co., Ltd., Jinan, China) and were not corrected. The ^1^H NMR and ^13^C NMR spectra were determined on a Bruker Avance III HD 500 MHz/600 MHz spectrometer (Switzerland Bruker Co., Ltd., Zurich, Switzerland) using CDCl_3_ as the solvent and TMS as an internal standard. Mass spectra were obtained by means of the electrospray ionization (ESI) method on the TSQ Quantum Access MAX HPLC-MS instrument (Thermo Scientific Co., Ltd., Waltham, MA, USA). Elemental analyses were measured using a PE 2400 II elemental analyzer (Perkin-Elmer Instruments Co., Ltd., Waltham, MA, USA). L-menthol (GC purity 99%) was provided by Shanghai Macklin Biochemical Co., Ltd. (Shanghai, China). A series of substituted aryl hydrazides were purchased from Shanghai Aladdin Biochemical Technology Co., Ltd. (Shanghai, China). Other reagents were purchased from commercial suppliers and used as received.

### 3.2. Synthesis of (-)-Menthyl-2-chloroacetate *(**2**)*

A solution of chloroacetyl chloride (12.42 g, 0.11 mol) in dry DCM (20 mL) was added slowly to a mixture of (-)-menthol (15.63 g, 0.10 mol), triethylamine (11.13 g, 0.11 mol) and DMAP (0.80 g, 0.006 mol) in dry DCM (15 mL) with ice-bath cooling. The reaction process was monitored by TLC. Upon completion, saturated aqueous NaHCO_3_ (5 mL) was added to destroy the unreacted acyl chloride. Then, the organic layer was separated, washed with deionized water three times, dried over Na_2_SO_4_, and concentrated under reduced pressure. The crude product was further purified by silica gel chromatography (petroleum ether/ethyl acetate 10:1, v/v) to obtain the intermediate **2** as a colorless transparent liquid. Yield 92%; ^1^H NMR (600 MHz, CDCl_3_) δ 4.73 (td, *J* = 10.9, 4.4 Hz, 1H), 4.10 (s, 2H), 2.00 (dt, *J* = 12.1, 4.1 Hz, 1H), 1.85 (pt, *J* = 7.0, 3.5 Hz, 1H), 1.69 (dt, *J* = 13.9, 3.4 Hz, 2H), 1.48 (dtd, *J* = 12.1, 6.7, 6.1, 3.2 Hz, 1H), 1.40 (s, 1H), 1.07–0.98 (m, 2H), 0.92–0.88 (m, 6H), 0.85 (dd, *J* = 12.3, 3.6 Hz, 1H), 0.75 (d, *J* = 7.0 Hz, 3H); ^13^C NMR (151 MHz, CDCl_3_) δ 168.07, 76.35, 46.93, 40.63, 36.03, 34.11, 31.41, 26.18, 23.36, 21.97, 20.75, 16.28. These data were in agreement with that of the literature [39].

### 3.3. General Procedure of Menthol-Based 1,2,4-Triazole-thioether Compounds *(**5a**–**5w**)*

Intermediate **4** was prepared according to a method described in the literature [40]. The intermediate **4** (2.0 mmol) and sodium acetate trihydrate (0.27 g, 2.0 mmol) were dissolved in 15 mL mixed solvent of ethanol and water (ethanol/water 2:1, *v*/*v*). Then, the mixture was stirred at 45 °C for 2 h. Afterwards, a solution of menthyl chloroacetate 2 (0.46 g, 2.0 mmol) in 5 mL EtOH was slowly added to the mixture and continuously refluxed at 80 °C for 6 h. Upon completion of the reaction, the mixture was concentrated in vacuum. Then, the crude product was poured into saturated sodium bicarbonate solution, and the mixture was extracted with dichloromethane (3 × 20 mL). The organic layer was washed with saturated NaCl solution three times, dried over anhydrous sodium sulfate, and purified by silica gel column chromatography (petroleum ether/ethyl acetate 3:1, *v*/*v*) to afford the target compounds 5a–5w in yields of 50% to 76%.

Compound (**5a**): *(1R,2S,5R)-2-isopropyl-5-methylcyclohexyl-2-((4-methyl-5-phenyl-4H-1,2,4-triazol-3-yl) thio)acetate.* White solid. Yield: 63.0%, m.p. 124.7–125.1 °C; IR (KBr, cm^−1^): 2960, 2934, 2915, 2864 (s, C-H), 1735(s, C=O), 1482 (s, C=N), 703 (w, C-S-C); ^1^H NMR (600 MHz, CDCl_3_) δ 7.62 (dd, *J* = 6.6, 3.0 Hz, 2H, C_16_-H, C_20_-H), 7.54–7.46 (m, 3H, C_17_-H, C_18_-H, C_19_-H), 4.71 (td, *J* = 10.9, 4.5 Hz, 1H, C_1_-H), 4.08 (s, 2H, C_12_-H), 3.66 (s, 3H, C_21_-H), 1.98 (d, *J* = 12.6 Hz, 1H, C_6_-Ha), 1.86–1.79 (m, 1H, C_7_-H), 1.69–1.64 (m, 2H, C_4_-Ha, C_3_-Ha), 1.45 (ddt, *J* = 8.8, 5.6, 2.6 Hz, 1H, C_5_-H), 1.38 (t, *J* = 11.7 Hz, 1H, C_2_-H), 1.06–0.95 (m, 2H, C_6_-H_b_, C_3_-H_b_), 0.88 (dd, *J* = 8.7, 6.7 Hz, 6H, C_10_-H, C_8_-H or C_9_-H), 0.83 (dd, *J* = 12.7, 3.4 Hz, 1H, C_4_-H_b_), 0.73 (d, *J* = 7.0 Hz, 3H, C_8_-H or C_9_-H); ^13^C NMR (150 MHz, CDCl_3_) δ 168.07, 156.10, 150.54, 130.15, 128.92, 128.62, 126.99, 76.35, 46.93, 40.63, 36.03, 34.11, 31.86, 31.41, 26.18, 23.36, 21.97, 20.75, 16.28; ESI-MS *m*/*z*: 388.20 [M + H]^+^. Anal. Calcd. for C_21_H_29_N_3_O_2_S: C, 65.08; H, 7.54; N,10.84; Found: C, 65.06; H, 7.52; N, 10.37. 

Compound (**5b**): *(1R,2S,5R)-2-isopropyl-5-methylcyclohexyl-2-((4-methyl-5-(o-tolyl)-4H-1,2,4-triazol-3-yl)thio)acetate.* Pale yellow liquid. Yield: 58.3%, IR (KBr, cm^−1^): 2954, 2927, 2869 (s, C-H), 1731(s, C=O), 1457 (s, C=N), 706 (w, C-S-C); ^1^H NMR (500 MHz, CDCl_3_) δ 7.42 (ddd, *J* = 8.5, 6.0, 2.7 Hz, 1H, C_20_-H), 7.35 (d, *J* = 7.6 Hz, 1H, C_19_-H), 7.32–7.29 (m, 2H, C_18_-H, C_17_-H), 4.75 (td, *J* = 10.9, 4.4 Hz, 1H, C_1_-H), 4.12 (s, 2H, C_12_-H), 3.44 (s, 3H, C_21_-H), 2.25 (s, 3H, C_22_-H), 2.01 (dtd, *J* = 12.0, 4.1, 1.8 Hz, 1H, C_6_-Ha), 1.90 (d, *J* = 4.1 Hz, 1H, C_7_-H), 1.71 (t, *J* = 2.6 Hz, 1H, C_4_-Ha), 1.68 (q, *J* = 2.8 Hz, 1H, C_3_-Ha), 1.49 (dddd, *J* = 12.1, 8.9, 6.3, 3.2 Hz, 1H, C_5_-H), 1.45–1.38 (m, 1H, C_2_-H), 1.11–1.04 (m, 1H, C_3_-H_b_), 1.04–0.99 (m, 1H, C_6_-H_b_), 0.92 (d, *J* = 4.6 Hz, 3H, C_10_-H), 0.91 (d, *J* = 5.1 Hz, 3H, C_8_-H or C_9_-H), 0.88 (d, *J* = 9.6 Hz, 1H, C_4_-H_b_), 0.76 (d, *J* = 7.0 Hz, 3H, C_8_-H or C_9_-H); ^13^C NMR (125 MHz, CDCl_3_) δ 168.21, 155.91, 149.79, 138.45, 130.79, 130.54, 130.34, 126.57, 126.11, 76.47, 47.05, 40.76, 36.05, 34.23, 31.53, 31.00, 26.32, 23.49, 22.09, 20.86, 19.82, 16.41; ESI-MS *m*/*z*: 402.22 [M + H]^+^. Anal. Calcd. for C_22_H_31_N_3_O_2_S: C, 65.80; H, 7.78; N,10.46; Found: C, 65.72; H, 7.73; N, 10.40.

Compound (**5c**): *(1R,2S,5R)-2-isopropyl-5-methylcyclohexyl-2-((4-methyl-5-(p-tolyl)-4H-1,2,4-triazol-3-yl)thio)acetate.* White solid. Yield: 70.0%, m.p. 124.0–125.2 °C; IR (KBr, cm^−1^): 2947, 2935, 2917, 2865(s, C-H), 1735(s, C=O), 1475 (s, C=N), 707 (w, C-S-C); ^1^H NMR (500 MHz, CDCl_3_) δ 7.51 (d, *J* = 8.2 Hz, 2H, C_15_-H, C_20_-H), 7.30 (d, *J* = 7.9 Hz, 2H, C_17_-H, C_19_-H), 4.71 (td, *J* = 10.9, 4.4 Hz, 1H, C_1_-H), 4.07 (s, 2H, C_12_-H), 3.65 (s, 3H, C_21_-H), 2.42 (s, 3H, C_22_-H), 1.98 (dtd, *J* = 12.1, 4.1, 1.8 Hz, 1H, C_6_-Ha), 1.82 (dt, *J* = 6.9, 3.5 Hz, 1H, C_7_-H), 1.68 (h, *J* = 2.1 Hz, 1H, C_4_-Ha), 1.65 (q, *J* = 2.9 Hz, 1H, C_3_-Ha), 1.45 (dddt, *J* = 12.3, 9.6, 6.6, 3.3 Hz, 1H, C_5_-H), 1.38 (ddt, *J* = 14.1, 10.9, 3.2 Hz, 1H, C_2_-H), 1.08–1.01 (m, 1H, C_3_-H_b_), 1.01–0.95 (m, 1H, C_6_-H_b_), 0.89 (d, *J* = 6.6 Hz, 3H, C_10_-H), 0.87 (d, *J* = 7.1 Hz, 3H, C_8_-H or C_9_-H), 0.86–0.81 (m, 1H, C_4_-H_b_), 0.72 (d, *J* = 6.9 Hz, 3H, C_8_-H or C_9_-H); ^13^C NMR (125 MHz, CDCl_3_) δ 168.11, 156.18, 150.29, 140.35, 129.59, 128.49, 124.06, 76.33, 46.92, 40.63, 36.06, 34.11, 31.83, 31.41, 26.17, 23.36, 21.97, 21.44, 20.75, 16.27; ESI-MS *m*/*z*: 402.21 [M + H]^+^. Anal. Calcd. for C_22_H_31_N_3_O_2_S: C, 65.80; H, 7.78; N,10.46; Found: C, 65.70; H, 7.72; N, 10.42.

Compound (**5d**): *(1R,2S,5R)-2-isopropyl-5-methylcyclohexyl-2-((5-(2-methoxyphenyl)-4-methyl-4H-1,2,4-triazol-3-yl)thio)acetate*. Pale yellow liquid. Yield: 68.2%, IR (KBr, cm^−1^): 2954, 2928, 2869 (s, C-H), 1731(s, C=O), 1476 (s, C=N), 700 (w, C-S-C); ^1^H NMR (500 MHz, CDCl_3_) δ 7.51–7.45 (m, 2H, C_20_-H, C_18_-H), 7.07 (t, *J* = 7.5 Hz, 1H, C_19_-H), 6.99 (d, *J* = 8.7 Hz, 1H, C_17_-H), 4.72 (td, *J* = 10.9, 4.4 Hz, 1H, C_1_-H), 4.10 (s, 2H, C_12_-H), 3.81 (s, 3H, C_22_-H), 3.44 (s, 3H, C_21_-H), 2.02–1.96 (m, 1H, C_6_-Ha), 1.86 (dd, *J* = 7.0, 4.2 Hz, 1H, C_7_-H), 1.68 (q, *J* = 2.8, 2.2 Hz, 1H, C_4_-Ha), 1.65 (q, *J* = 2.8 Hz, 1H, C_3_-Ha), 1.47 (dddq, *J* = 15.3, 8.9, 6.2, 3.3 Hz, 1H, C_5_-H), 1.42–1.35 (m, 1H, C_2_-H), 1.09–1.02 (m, 1H, C_3_-H_b_), 1.01–0.96 (m, 1H, C_6_-H_b_), 0.89 (t, *J* = 6.6 Hz, 6H, C_10_-H, C_8_-H or C_9_-H), 0.83 (m, 1H, C_4_-H_b_), 0.74 (d, *J* = 7.0 Hz, 3H, C_8_-H or C_9_-H); ^13^C NMR (125 MHz, CDCl_3_) δ 168.14, 157.23, 154.60, 149.85, 132.27, 132.15, 121.12, 116.15, 111.07, 76.30, 55.55, 46.93, 40.63, 35.86, 34.13, 31.42, 31.16, 26.18, 23.38, 21.98, 20.74, 16.28; ESI-MS *m*/*z*: 418.22 [M + H]^+^. Anal. Calcd. for C_22_H_31_N_3_O_3_S: C, 63.28; H, 7.48; N,10.06; Found: C, 63.25; H, 7.45; N, 10.05.

Compound (**5e**): *(1R,2S,5R)-2-isopropyl-5-methylcyclohexyl-2-((5-(4-methoxyphenyl)-4-methyl-4H-1,2,4-triazol-3-yl)thio)acetate.* White solid. Yield: 66.8%, m.p. 106.9–108.4 °C; IR (KBr, cm^−1^): 2952, 2936, 2869 (s, C-H), 1735(s, C=O), 1477 (s, C=N), 714 (w, C-S-C); ^1^H NMR (500 MHz, CDCl_3_) δ 7.56 (d, J = 8.8 Hz, 2H, C_16_-H, C_20_-H), 7.00 (d, J = 8.7 Hz, 2H, C_17_-H, C_19_-H), 4.70 (td, *J* = 10.9, 4.4 Hz, 1H, C_1_-H), 4.06 (s, 2H, C_12_-H), 3.85 (s, 3H, C_22_-H), 3.64 (s, 3H, C_21_-H), 2.00–1.95 (m, 1H, C_6_-Ha), 1.82 (qt, *J* = 7.0, 3.5 Hz, 1H, C_7_-H), 1.67 (q, *J* = 2.5, 1.8 Hz, 1H, C_4_-Ha), 1.64 (q, *J* = 2.9 Hz, 1H, C_3_-Ha), 1.46 (dtd, *J* = 12.1, 5.9, 5.5, 3.0 Hz, 1H, C_5_-H), 1.42–1.35 (m, 1H, C_2_-H), 1.08–1.00 (m, 1H, C_3_-H_b_), 1.00–0.93 (m, 1H, C_6_-H_b_), 0.88 (d, *J* = 6.8 Hz, 3H, C_10_-H), 0.87 (d, *J* = 7.7 Hz, 3H, C_8_-H or C_9_-H), 0.85–0.80 (m, 1H, C_4_-H_b_), 0.72 (d, *J* = 6.9 Hz, 3H, C_8_-H or C_9_-H); ^13^C NMR (125 MHz, CDCl_3_) δ 168.13, 161.01, 156.00, 150.12, 130.06, 119.26, 114.36, 76.33, 55.41, 46.92, 40.63, 36.09, 34.11, 31.83, 31.41, 26.17, 23.35, 21.97, 20.75, 16.27; ESI-MS *m*/*z*: 418.23 [M + H]^+^. Anal. Calcd. for C_22_H_31_N_3_O_3_S: C, 63.28; H, 7.48; N,10.06; Found: C, 63.23; H, 7.43; N, 10.02.

Compound (**5f**): *(1R,2S,5R)-2-isopropyl-5-methylcyclohexyl-2-((5-(2-fluorophenyl)-4-methyl-4H-1,2,4-triazol-3-yl)thio) acetate.* Pale yellow liquid. Yield: 62.8%, IR (KBr, cm^−1^): 2955, 2928, 2869 (s, C-H), 1732(s, C=O), 1474 (s, C=N), 671 (w, C-S-C); ^1^H NMR (500 MHz, CDCl_3_) δ 7.63 (td, *J* = 7.4, 1.8 Hz, 1H, C_20_-H), 7.52 (tdd, *J* = 7.5, 5.3, 1.8 Hz, 1H, C_18_-H), 7.30 (t, *J* = 7.5 Hz, 1H, C_19_-H), 7.24–7.18 (m, 1H, C_17_-H), 4.72 (td, *J* = 10.9, 4.4 Hz, 1H, C_1_-H), 4.10 (s, 2H, C_12_-H), 3.55 (d, *J* = 2.4 Hz, 3H, C_21_-H), 1.99 (d, *J* = 12.0 Hz, 1H, C_6_-H_a_), 1.85 (qd, *J* = 7.0, 2.7 Hz, 1H, C_7_-H), 1.68 (q, *J* = 2.8, 2.3 Hz, 1H, C_4_-H_a_), 1.65 (q, *J* = 2.8 Hz, 1H, C_3_-H_a_), 1.46 (dddq, *J* = 12.3, 9.5, 6.5, 3.3 Hz, 1H, C_5_-H), 1.38 (ddt, *J* = 14.2, 11.0, 3.1 Hz, 1H, C_2_-H), 1.08–1.01 (m, 1H, C_3_-H_b_), 1.01–0.95 (m, 1H, C_6_-H_b_), 0.88 (t, *J* = 7.1 Hz, 6H, C_10_-H, C_8_-H or C_9_-H), 0.83 (dd, *J* = 12.3, 3.4 Hz, 1H, C_4_-H_b_), 0.73 (d, *J* = 6.9 Hz, 3H, C_8_-H or C_9_-H); ^13^C NMR (125 MHz, CDCl_3_) δ 167.99, 159.72 (d, *J* = 249.9 Hz), 152.21, 150.82, 132.67 (d, *J* = 8.2 Hz), 132.27, 124.97 (d, *J* = 3.6 Hz), 116.07 (d, *J* = 21.3 Hz), 115.18 (d, *J* = 14.6 Hz), 76.40, 46.92, 40.61, 35.84, 34.12, 31.41, 31.32 (d, *J* = 6.2 Hz), 26.19, 23.37, 21.96, 20.73, 16.26; ESI-MS *m*/*z*: 406.20 [M + H]^+^. Anal. Calcd. for C_21_H_28_FN_3_O_2_S: C, 62.20; H, 6.96; N,10.36; Found: C, 61.18; H, 6.94; N, 10.32.

Compound (**5g**): *(1R,2S,5R)-2-isopropyl-5-methylcyclohexyl-2-((5-(3-fluorophenyl)-4-methyl-4H-1,2,4-triazol-3-yl)thio) acetate.* Pale yellow liquid. Yield: 65.3%, IR (KBr, cm^−1^): 2959, 2933, 2868 (s, C-H), 1736(s, C=O), 1484 (s, C=N), 683 (w, C-S-C); ^1^H NMR (500 MHz, CDCl_3_) δ 7.46 (td, *J* = 7.9, 5.6 Hz, 1H, C_19_-H), 7.42–7.38 (m, 1H, C_20_-H), 7.35 (dt, *J* = 9.3, 2.1 Hz, 1H, C_16_-H), 7.17 (td, *J* = 8.9, 8.4, 2.5 Hz, 1H, C_18_-H), 4.69 (td, *J* = 10.9, 4.4 Hz, 1H, C_1_-H), 4.06 (s, 2H, C_12_-H), 3.66 (s, 3H, C_21_-H), 1.95 (dtd, *J* = 12.1, 4.0, 1.7 Hz, 1H, C_6_-H_a_), 1.80 (pd, *J* = 7.0, 2.7 Hz, 1H, C_7_-H), 1.66 (q, *J* = 2.6, 1.9 Hz, 1H, C_4_-H_a_), 1.64–1.61 (m, 1H, C_3_-H_a_), 1.44 (dddt, *J* = 12.1, 8.8, 6.2, 3.4 Hz, 1H, C_5_-H), 1.36 (ddt, *J* = 14.2, 10.9, 3.2 Hz, 1H, C_2_-H), 1.06–0.98 (m, 1H, C_3_-H_b_), 0.99–0.93 (m, 1H, C_6_-H_b_), 0.88–0.84 (m, 6H, C_10_-H, C_8_-H or C_9_-H), 0.83– 0.78 (m, 1H, C_4_-H_b_), 0.70 (d, *J* = 7.0 Hz, 3H, C_8_-H or C_9_-H); ^13^C NMR (125 MHz, CDCl_3_) δ 168.11, 162.85(d, *J* = 248.0 Hz), 155.02, 151.17, 130.82 (d, *J* = 8.2 Hz), 129.03 (d, *J* = 8.5 Hz), 124.35 (d, *J* = 3.2 Hz), 117.36 (d, *J* = 21.0 Hz), 115.86 (d, *J* = 23.3 Hz), 76.55, 47.05, 40.76, 36.08, 34.23, 32.06, 31.53, 26.31, 23.48, 22.08, 20.86, 16.40; ESI-MS *m*/*z*: 406.20 [M + H]^+^. Anal. Calcd. for C_21_H_28_FN_3_O_2_S: C, 62.20; H, 6.96; N,10.36; Found: C, 61.17; H, 6.93; N, 10.31.

Compound (**5h**): *(1R,2S,5R)-2-isopropyl-5-methylcyclohexyl-2-((5-(4-fluorophenyl)-4-methyl-4H-1,2,4-triazol-3-yl)thio) acetate.* White solid. Yield: 60.1%, m.p. 136.5–137.9 °C; IR (KBr, cm^−1^): 2958, 2929, 2869 (s, C-H), 1732 (s, C=O), 1484 (s, C=N), 689 (w, C-S-C). ^1^H NMR (500 MHz, CDCl_3_) δ 7.62 (dd, *J* = 8.8, 5.2 Hz, 2H, C_20_-H, C_16_-H), 7.20 (t, *J* = 8.6 Hz, 2H, C_19_-H, C_17_-H), 4.71 (td, *J* = 10.9, 4.4 Hz, 1H, C_1_-H), 4.07 (s, 2H, C_12_-H), 3.65 (s, 3H, C_21_-H), 1.98 (dtd, *J* = 12.0, 4.0, 1.7 Hz, 1H, C_6_-H_a_), 1.83 (qd, *J* = 7.0, 2.8 Hz, 1H, C_7_-H), 1.68 (q, *J* = 2.6, 1.9 Hz, 1H, C_4_-H_a_), 1.65 (q, *J* = 2.8 Hz, 1H, C_3_-H_a_), 1.46 (dddd, *J* = 12.0, 8.9, 6.4, 3.3 Hz, 1H, C_5_-H), 1.38 (ddt, *J* = 14.2, 10.8, 3.1 Hz, 1H, C_2_-H), 1.07–1.00 (m, 1H, C_3_-H_b_), 0.99–0.95 (m, 1H, C_6_-H_b_), 0.90–0.86 (m, 6H, C_10_-H, C_8_-H or C_9_-H), 0.86–0.81 (m, 1H, C_4_-H_b_), 0.72 (d, *J* = 6.9 Hz, 3H, C_8_-H or C_9_-H); ^13^C NMR (125 MHz, CDCl_3_) δ 168.17, 163.92 (d, J = 251.1 Hz), 155.35, 150.81, 130.80 (d, *J* = 8.6 Hz), 123.27 (d, *J* = 3.6 Hz), 116.32 (d, *J* = 22.1 Hz), 76.53, 47.05, 40.76, 36.12, 34.23, 31.97, 31.54, 26.31, 23.49, 22.09, 20.87, 16.41; ESI-MS *m*/*z*: 406.19 [M + H]^+^. Anal. Calcd. for C_21_H_28_FN_3_O_2_S: C, 62.20; H, 6.96; N,10.36; Found: C, 61.17; H, 6.92; N, 10.30.

Compound (**5i**): *(1R,2S,5R)-2-isopropyl-5-methylcyclohexyl-2-((5-(2-chlorophenyl)-4-methyl-4H-1,2,4-triazol-3-yl)thio)acetate.* Pale yellow liquid. Yield: 60.8%; IR (KBr, cm^−1^): 2954, 2927, 2869 (s, C-H), 1731 (s, C=O), 1467 (s, C=N), 698 (w, C-S-C). ^1^H NMR (500 MHz, CDCl_3_) δ 7.54–7.44 (m, 3H, Ar-H), 7.42- 7.36 (m, 1H, Ar-H), 4.73 (td, *J* = 10.9, 4.4 Hz, 1H, C_1_-H), 4.11 (s, 2H, C_12_-H), 3.48 (s, 3H, C_21_-H), 1.98 (dd, *J* = 10.7, 5.9 Hz, 1H, C_6_-H_a_), 1.87 (dt, *J* = 7.1, 3.5 Hz, 1H, C_7_-H), 1.68 (q, *J* = 2.8, 2.4 Hz, 1H, C_4_-H_a_), 1.65 (q, *J* = 2.8 Hz, 1H, C_3_-H_a_), 1.46 (ddt, *J* = 8.9, 5.7, 2.8 Hz, 1H, C_5_-H), 1.42–1.35 (m, 1H, C_2_-H), 1.05 (ddd, *J* = 13.1, 9.7, 3.1 Hz, 1H, C_3_-H_b_), 1.01–0.95 (m, 1H, C_6_-H_b_), 0.91–0.87 (m, 6H, C_10_-H, C_8_-H or C_9_-H), 0.87–0.82 (m, 1H, C_4_-H_b_), 0.74 (d, *J* = 7.0 Hz, 3H, C_8_-H or C_9_-H); ^13^C NMR (125 MHz, CDCl3) δ 167.99, 154.22, 150.37, 134.23, 132.65, 131.97, 129.89, 127.26, 126.58, 76.39, 46.93, 40.63, 35.85, 34.12, 31.41, 31.11, 26.20, 23.37, 21.98, 20.75, 16.29; ESI-MS *m*/*z*: C 422.17 [M + H]^+^. Anal. Calcd. for C_21_H_28_ClN_3_O_2_S: C, 59.77; H, 6.69; N,9.96; Found: C, 59.76; H, 6.67; N, 9.93.

Compound (**5j**): *(1R,2S,5R)-2-isopropyl-5-methylcyclohexyl-2-((5-(3-chlorophenyl)-4-methyl-4H-1,2,4-triazol-3-yl)thio)acetate.* White solid. Yield: 72.1%, m.p. 68.4–70.0 °C; IR (KBr, cm^−1^): 2955, 2935, 2869 (s, C-H), 1732 (s, C=O), 1471 (s, C=N), 685 (s, C-S-C). ^1^H NMR (500 MHz, CDCl_3_) δ 7.64 (d, *J* = 2.0 Hz, 1H, C_20_-H), 7.52 (dt, *J* = 7.4, 1.7 Hz, 1H, C_16_-H), 7.48 (dt, *J* = 8.1, 1.7 Hz, 1H, C_19_-H), 7.46–7.42 (m, 1H, C_18_-H), 4.71 (td, *J* = 10.9, 4.4 Hz, 1H, C_1_-H), 4.08 (s, 2H, C_12_-H), 3.67 (s, 3H, C_21_-H), 1.97 (dtd, *J* = 12.1, 4.1, 1.7 Hz, 1H, C_6_-H_a_), 1.82 (qt, *J* = 7.0, 3.5 Hz, 1H, C_7_-H), 1.68 (q, *J* = 2.6, 1.9 Hz, 1H, C_4_-H_a_), 1.65 (q, *J* = 2.9 Hz, 1H, C_3_-H_a_), 1.46 (dddt, *J* = 12.1, 8.9, 6.1, 3.3 Hz, 1H, C_5_-H), 1.38 (ddt, *J* = 14.2, 10.9, 3.1 Hz, 1H, C_2_-H), 1.10–1.01 (m, 1H, C_3_-H_b_), 1.01–0.95 (m, 1H, C_6_-H_b_), 0.88 (t, *J* = 7.2 Hz, 6H, C_10_-H, C_8_-H or C_9_-H), 0.86–0.81 (m, 1H, C_4_-H_b_), 0.73 (d, *J* = 7.0 Hz, 3H, C_8_-H or C_9_-H); ^13^C NMR (125 MHz, CDCl_3_) δ 167.98, 154.79, 151.06, 134.99, 130.30, 130.25, 128.66, 128.63, 126.65, 76.43, 46.93, 40.64, 35.98, 34.11, 31.93, 31.41, 26.19, 23.36, 21.97, 20.75, 16.29; ESI-MS *m*/*z*: 422.18 [M + H]^+^. Anal. Calcd. for C_21_H_28_ClN_3_O_2_S: C, 59.77; H, 6.69; N,9.96; Found: C, 59.75; H, 6.65; N, 9.92.

Compound (**5k**): *(1R,2S,5R)-2-isopropyl-5-methylcyclohexyl-2-((5-(2-chlorophenyl)-4-methyl-4H-1,2,4-triazol-3-yl)thio)acetate.* White solid. Yield: 66.2%, m.p. 121.9–123.7 °C; IR (KBr, v/cm^−1^): 2956, 2932, 2869 (s, C-H), 1732(s, C=O), 1478 (s, C=N), 713 (w, C-S-C). ^1^H NMR (500 MHz, CDCl_3_) δ 7.58 (d, *J* = 8.5 Hz, 2H, C_16_-H, C_20_-H), 7.48 (d, *J* = 8.6 Hz, 2H, C_17_-H, C_19_-H), 4.71 (td, *J* = 10.9, 4.4 Hz, 1H, C_1_-H), 4.07 (s, 2H, C_12_-H), 3.66 (s, 3H, C_21_-H), 1.97 (dtd, *J* = 12.1, 4.1, 1.8 Hz, 1H, C_6_-H_a_), 1.81 (qd, *J* = 7.0, 2.7 Hz, 1H, C_7_-H), 1.68 (q, *J* = 2.6, 1.9 Hz, 1H, C_4_-H_a_), 1.65 (q, *J* = 2.9 Hz, 1H, C_3_-H_a_), 1.46 (dddt, *J* = 12.0, 8.9, 6.2, 3.4 Hz, 1H, C_5_-H), 1.41–1.35 (m, 1H, C_2_-H), 1.09–1.01 (m, 1H, C_3_-H_b_), 1.00–0.95 (m, 1H, C_6_-H_b_), 0.88 (dd, *J* = 8.5, 6.7 Hz, 6H, C_10_-H, C_8_-H or C_9_-H), 0.86–0.79 (m, 1H, C_4_-H_b_), 0.72 (d, *J* = 7.0 Hz, 3H, C_8_-H or C_9_-H); ^13^C NMR (125 MHz, CDCl_3_) δ 168.01, 155.09, 150.92, 136.47, 129.83, 129.28, 125.43, 76.42, 46.93, 40.63, 35.98, 34.10, 31.90, 31.41, 26.18, 23.36, 21.97, 20.74, 16.28; ESI-MS *m*/*z*: 422.17 [M + H]^+^. Anal. Calcd. for C_21_H_28_ClN_3_O_2_S: C, 59.77; H, 6.69; N,9.96; Found: C, 59.74; H, 6.63; N, 9.93.

Compound (**5l**): *(1R,2S,5R)-2-isopropyl-5-methylcyclohexyl-2-((5-(4-bromophenyl)-4-methyl-4H-1,2,4-triazol-3-yl)thio)acetate.* White solid. Yield: 67.1%, m.p. 144.4–145.2 °C; IR (KBr, cm^−1^): 2955, 2931, 2870 (s, C-H), 1731 (s, C=O), 1477 (s, C=N), 713 (w, C-S-C). ^1^H NMR (500 MHz, CDCl_3_) δ 7.68–7.62 (m, 2H, C_16_-H, C_20_-H), 7.54–7.48 (m, 2H, C_17_-H, C_19_-H), 4.71 (td, *J* = 10.9, 4.4 Hz, 1H, C_1_-H), 4.08 (s, 2H, C_12_-H), 3.66 (s, 3H, C_21_-H), 1.97 (dtd, *J* = 12.1, 4.2, 1.8 Hz, 1H, C_6_-Ha), 1.82 (pd, *J* = 7.0, 2.8 Hz, 1H, C_7_-H), 1.68 (q, *J* = 2.6, 1.9 Hz, 1H, C_4_-Ha), 1.65 (q, *J* = 2.9 Hz, 1H, C_3_-Ha), 1.46 (dddd, *J* = 12.1, 8.9, 6.2, 3.2 Hz, 1H, C_5_-H), 1.38 (ddt, *J* = 14.2, 10.9, 3.1 Hz, 1H, C_2_-H), 1.08–1.01 (m, 1H, C_3_-H_b_), 1.00–0.95 (m, 1H, C_6_-H_b_), 0.88 (dd, *J* = 8.7, 6.7 Hz, 6H, C_10_-H, C_8_-H or C_9_-H), 0.86–0.81 (m, 1H, C_4_-H_b_), 0.72 (d, *J* = 6.9 Hz, 3H, C_8_-H or C_9_-H); ^13^C NMR (125 MHz, CDCl_3_) δ 168.00, 155.13, 150.97, 132.24, 130.02, 125.88, 124.74, 76.42, 46.92, 40.63, 35.97, 34.10, 31.91, 31.41, 26.18, 23.35, 21.97, 20.74, 16.28; ESI-MS *m*/*z*: 466.13 [M + H]^+^. Anal. Calcd. for C_21_H_28_BrN_3_O_2_S: C, 54.08; H, 6.05; N,9.01; Found: C, 54.06; H, 6.02; N, 9.00.

Compound (**5m**): *(1R,2S,5R)-2-isopropyl-5-methylcyclohexyl-2-((5-(2-iodophenyl)-4-methyl-4H-1,2,4-triazol-3-yl)thio)acetate.* Pale yellow liquid. Yield: 50.6%; IR (KBr, cm^−1^): 2954, 2927, 2868 (s, C-H), 1731 (s, C=O), 1466 (s, C=N), 692 (w, C-S-C); ^1^H NMR (500 MHz, CDCl_3_) δ 7.95 (d, *J* = 8.0 Hz, 1H, C_17_-H), 7.50–7.46 (m, 1H, C_18_-H), 7.38 (dd, *J* = 7.6, 1.7 Hz, 1H, C_20_-H), 7.22 (td, *J* = 7.7, 1.8 Hz, 1H, C_19_-H), 4.73 (td, *J* = 11.0, 4.4 Hz, 1H, C_1_-H), 4.12 (s, 2H, C_12_-H), 3.44 (s, 3H, C_21_-H), 2.00 (dtd, *J* = 12.0, 4.0, 1.8 Hz, 1H, C_6_-H_a_), 1.88 (pd, *J* = 7.0, 2.7 Hz, 1H, C_7_-H), 1.69 (q, *J* = 2.8 Hz, 1H, C_4_-H_a_), 1.66 (h, *J* = 2.8 Hz, 1H, C_3_-H_a_), 1.47 (dddt, *J* = 15.3, 8.8, 6.2, 3.2 Hz, 1H, C_5_-H), 1.43–1.37 (m, 1H, C_2_-H), 1.10–1.03 (m, 1H, C_3_-H_b_), 1.03–0.97 (m, 1H, C_6_-H_b_), 0.90 (d, *J* = 3.4 Hz, 3H, C_10_-H), 0.89 (d, *J* = 3.9 Hz, 3H, C_8_-H or C_9_-H), 0.87–0.84 (m, 1H, C_4_-H_b_), 0.75 (d, *J* = 7.0 Hz, 3H, C_8_-H or C_9_-H); ^13^C NMR (125 MHz, CDCl_3_) δ 168.11, 157.37, 150.15, 139.51, 133.05, 132.13, 132.08, 128.53, 98.86, 76.52, 47.06, 40.79, 36.04, 34.25, 31.54, 31.39, 26.32, 23.50, 22.12, 20.90, 16.45; ESI-MS *m*/*z*: 514.13 [M + H]^+^. Anal. Calcd. for C_21_H_28_IN_3_O_2_S: C, 49.13; H, 5.50; N, 8.18; Found: C, 49.11; H, 5.48; N, 8.15.

Compound (**5n**): *(1R,2S,5R)-2-isopropyl-5-methylcyclohexyl-2-((5-(4-iodophenyl)-4-methyl-4H-1,2,4-triazol-3-yl)thio) acetate.* White solid. Yield: 67.1%, m.p. 131.1–131.8 °C; IR (KBr, cm^−1^): 2954, 2929, 2869 (s, C-H), 1731 (s, C=O), 1475 (s, C=N), 710 (w, C-S-C); ^1^H NMR (500 MHz, CDCl_3_) δ 7.89–7.81 (m, 2H, C_17_-H, C_19_-H), 7.41–7.34 (m, 2H, C_16_-H, C_20_-H), 4.70 (td, *J* = 10.9, 4.4 Hz, 1H, C_1_-H), 4.07 (s, 2H, C_12_-H), 3.65 (s, 3H, C_21_-H), 1.97 (dtd, *J* = 12.0, 4.0, 1.7 Hz, 1H, C_6_-H_a_), 1.82 (pd, *J* = 7.1, 2.9 Hz, 1H, C_7_-H), 1.68 (q, *J* = 2.5, 1.9 Hz, 1H, C_4_-H_a_), 1.66–1.63 (m, 1H, C_3_-H_a_), 1.45 (dddt, *J* = 12.2, 9.5, 6.5, 3.2 Hz, 1H, C_5_-H), 1.38 (ddt, *J* = 14.1, 10.9, 3.2 Hz, 1H, C_2_-H), 1.08–1.00 (m, 1H, C_3_-H_b_), 1.00–0.95 (m, 1H, C_6_-H_b_), 0.89 (d, *J* = 6.5 Hz, 3H, C_10_-H), 0.87 (d, *J* = 7.0 Hz, 3H, C_8_-H or C_9_-H), 0.86–0.81 (m, 1H, C_4_-H_b_), 0.72 (d, *J* = 7.0 Hz, 3H, C_8_-H or C_9_-H); ^13^C NMR (125 MHz, CDCl_3_) δ 168.00, 155.24, 151.00, 138.17, 130.04, 126.42, 96.62, 76.43, 46.93, 40.64, 35.97, 34.10, 31.91, 31.41, 26.18, 23.36, 21.97, 20.75, 16.28. ESI-MS *m*/*z*: 514.11 [M + H]^+^. Anal. Calcd. for C_21_H_28_IN_3_O_2_S: C, 49.13; H, 5.50; N,8.18; Found: C, 49.10; H, 5.49; N, 8.16.

Compound (**5o**): *(1R,2S,5R)-2-isopropyl-5-methylcyclohexyl-2-((4-methyl-5-(2-(trifluoromethyl)phenyl)-4H-1,2,4-triazol-3-yl)thio)acetate.* White solid. Yield: 56.1%, m.p. 88.6–90.2 °C; IR (KBr, cm^−1^): 2954, 2929, 2869 (s, C-H), 1731 (s, C=O), 1475 (s, C=N), 710 (w, C-S-C); ^1^H NMR (500 MHz, CDCl_3_) δ 7.88–7.80 (m, 1H, C_20_-H), 7.72–7.64 (m, 2H, C_17_-H, C_19_-H), 7.49–7.43 (m, 1H, C_18_-H), 4.73 (td, *J* = 10.9, 4.4 Hz, 1H, C_1_-H), 4.11 (s, 2H, C_12_-H), 3.38 (s, 3H, C_21_-H), 1.98 (dd, *J* = 10.6, 5.9 Hz, 1H, C_6_-H_a_), 1.88 (ddt, *J* = 14.0, 7.0, 3.5 Hz, 1H, C_7_-H), 1.69 (q, *J* = 2.9 Hz, 1H, C_4_-H_a_), 1.66 (q, *J* = 3.3, 2.6 Hz, 1H, C_3_-H_a_), 1.48 (dddq, *J* = 12.0, 8.9, 6.1, 3.2 Hz, 1H, C_5_-H), 1.40 (ddt, *J* = 13.8, 11.0, 3.1 Hz, 1H, C_2_-H), 1.09–1.02 (m, 1H, C_3_-H_b_), 1.02–0.97 (m, 1H, C_6_-H_b_), 0.89 (dd, *J* = 6.8, 4.9 Hz, 6H, C_10_-H, C_8_-H or C_9_-H), 0.84 (dd, *J* = 12.3, 3.4 Hz, 1H, C_4_-H_b_), 0.75 (d, *J* = 6.9 Hz, 3H, C_8_-H or C_9_-H); ^13^C NMR (125 MHz, CDCl_3_) δ 168.11, 153.42, 150.34, 132.54, 132.16, 131.06, 131.02, 127.21, 125.59, 124.57, 76.56, 47.04, 40.72, 36.06, 34.25, 31.55, 31.11, 26.3, 23.51, 22.07, 20.85, 16.39; ESI-MS *m*/*z*: 456.19 [M + H]^+^. Anal. Calcd. for C_22_H_28_F_3_N_3_O_2_S: C, 58.01; H, 6.20; N,9.22; Found: C, 58.00; H, 6.19; N, 9.21.

Compound (**5p**): *(1R,2S,5R)-2-isopropyl-5-methylcyclohexyl-2-((4-methyl-5-(3-(trifluoromethyl)phenyl)-4H-1,2,4-triazol-3-yl)thio)acetate.* White solid. Yield: 70.1%, m.p. 81.5–82.3 °C; IR (KBr, cm^−1^): 2961, 2928, 2869 (s, C-H), 1735 (s, C=O), 1459 (s, C=N), 704 (w, C-S-C); ^1^H NMR (500 MHz, CDCl_3_) δ 7.92 (d, *J* = 2.0 Hz, 1H, C_16_-H), 7.84 (d, *J* = 7.7 Hz, 1H, C_20_-H), 7.76 (d, *J* = 7.9 Hz, 1H, C_18_-H), 7.65 (t, *J* = 7.8 Hz, 1H, C_19_-H), 4.71 (td, *J* = 10.9, 4.4 Hz, 1H, C_1_-H), 4.09 (s, 2H, C_12_-H), 3.69 (s, 3H, C_21_-H), 1.98 (dtd, *J* = 12.0, 4.0, 1.7 Hz, 1H, C_6_-H_a_), 1.83 (qt, *J* = 7.0, 3.5 Hz, 1H, C_7_-H), 1.68 (q, *J* = 2.7, 2.0 Hz, 1H, C_4_-H_a_), 1.65 (q, *J* = 2.8 Hz, 1H, C_3_-H_a_), 1.46 (dddt, *J* = 12.1, 9.5, 6.6, 3.2 Hz, 1H, C_5_-H), 1.42–1.35 (m, 1H, C_2_-H), 1.08–1.01 (m, 1H, C_3_-H_b_), 1.01–0.96 (m, 1H, C_6_-H_b_), 0.88 (t, *J* = 6.6 Hz, 6H, C_10_-H, C_8_-H or C_9_-H), 0.86–0.81 (m, 1H, C_4_-H_b_), 0.73 (d, *J* = 7.0 Hz, 3H, C_8_-H or C_9_-H); ^13^C NMR (125 MHz, CDCl_3_) δ 168.09, 154.86, 151.45, 132.48, 129.74, 128.00, 127.00 (q, J = 3.8 Hz), 125.56 (q, *J* = 3.9 Hz), 125.15, 76.60, 47.06, 40.77, 36.11, 34.23, 32.06, 31.54, 26.33, 23.49, 22.08, 20.86, 16.41; ESI-MS *m/z:* 456.22 [M + H]^+^. Anal. Calcd. for C_22_H_28_F_3_N_3_O_2_S: C, 58.01; H, 6.20; N,9.22; Found: C, 57.98; H, 6.18; N, 9.20.

Compound (**5q**): *(1R,2S,5R)-2-isopropyl-5-methylcyclohexyl-2-((5-(2-hydroxyphenyl)-4-methyl-4H-1,2,4-triazol-3-yl)thio)acetate.* White solid. Yield: 65.1%, m.p. 158.9–160.9 °C; IR (KBr, cm^−1^): 3429(w, O-H), 2961, 2942, 2914, 2864 (s, C-H), 1735 (s, C=O), 1486 (s, C=N), 683 (w, C-S-C); ^1^H NMR (500 MHz, CDCl_3_) δ 11.18 (s, 1H, OH-H), 7.53 (dd, *J* = 8.0, 1.6 Hz, 1H, C_20_-H), 7.39–7.31 (m, 1H, C_18_-H), 7.12 (d, *J* = 8.2 Hz, 1H, C_19_-H), 6.95 (t, *J* = 7.6 Hz, 1H, C_17_-H), 4.71 (td, *J* = 10.9, 4.4 Hz, 1H, C_1_-H), 4.07 (s, 2H, C_12_-H), 3.86 (s, 3H, C_21_-H), 1.96 (dtd, *J* = 12.1, 4.2, 1.8 Hz, 1H, C_6_-H_a_), 1.81 (qt, *J* = 7.0, 3.5 Hz, 1H, C_7_-H), 1.67 (q, *J* = 2.6, 1.9 Hz, 1H, C_4_-H_a_), 1.64 (t, *J* = 2.9 Hz, 1H, C_3_-H_a_), 1.45 (dddt, *J* = 12.2, 9.4, 6.6, 3.1 Hz, 1H, C_5_-H), 1.37 (ddt, *J* = 14.2, 11.0, 3.1 Hz, 1H, C_2_-H), 1.07–1.00 (m, 1H, C_3_-H_b_), 1.00–0.94 (m, 1H, C_6_-H_b_), 0.88 (d, *J* = 6.6 Hz, 3H, C_10_-H), 0.86 (d, *J* = 7.0 Hz, 3H, C_8_-H or C_9_-H), 0.85–0.80 (m, 1H, C_4_-H_b_), 0.71 (d, *J* = 7.0 Hz, 3H, C_8_-H or C_9_-H); ^13^C NMR (125 MHz, CDCl_3_) δ 167.77, 157.57, 154.11, 151.18, 131.67, 125.49, 119.10, 118.15, 110.96, 76.54, 46.92, 40.63, 35.77, 34.08, 33.40, 31.40, 26.14, 23.31, 21.94), 20.72, 16.22; ESI-MS *m*/*z*: 404.20 [M + H]^+^. Anal. Calcd. for C_21_H_29_N_3_O_3_S: C, 62.50; H, 7.24; N,10.41; Found: C, 62.48; H, 7.20; N, 10.39.

Compound (**5r**): *(1R,2S,5R)-2-isopropyl-5-methylcyclohexyl-2-((5-(4-hydroxyphenyl)-4-methyl-4H-1,2,4-triazol-3-yl)thio)acetate.* White solid. Yield: 76.0%, m.p. 137.3–138.2 °C; IR (KBr, cm^−1^): 3435(w, O-H), 2953, 2929, 2866 (s, C-H), 1741 (s, C=O), 1476 (s, C=N), 698 (w, C-S-C); ^1^H NMR (500 MHz, CDCl_3_) δ 10.26 (s, 1H, OH-H), 7.35 (d, *J* = 8.6 Hz, 2H, C_16_-H, C_20_-H), 6.91 (d, *J* = 8.6 Hz, 2H, C_17_-H, C_19_-H), 4.71 (td, *J* = 10.9, 4.4 Hz, 1H, C_1_-H), 4.06 (s, 2H, C_12_-H), 3.62 (s, 3H, C_21_-H), 1.97 (d, *J* = 12.2 Hz, 1H, C_6_-H_a_), 1.81 (td, *J* = 7.0, 2.7 Hz, 1H, C_7_-H), 1.66 (d, *J* = 3.3 Hz, 1H, C_4_-H_a_), 1.63 (q, *J* = 2.8 Hz, 1H, C_3_-H_a_), 1.44 (ddt, *J* = 15.3, 8.8, 3.4 Hz, 1H, C_5_-H), 1.40–1.34 (m, 1H, C_2_-H), 1.07–1.00 (m, 1H, C_3_-H_b_), 1.00–0.93 (m, 1H, C_6_-H_b_), 0.86 (t, *J* = 6.7 Hz, 6H, C_10_-H, C_8_-H or C_9_-H), 0.81 (dd, *J* = 12.5, 3.4 Hz, 1H, C_4_-H_b_), 0.71 (d, *J* = 6.9 Hz, 3H, C_8_-H or C_9_-H); ^13^C NMR (125 MHz, CDCl_3_) δ 168.02, 159.76, 156.50, 150.24, 130.02, 116.50, 116.42, 76.49, 46.91, 40.61, 35.87, 34.09, 31.86, 31.40, 26.18, 23.34, 21.95, 20.72, 16.25; ESI-MS *m*/*z*: 404.19 [M + H]^+^. Anal. Calcd. for C_21_H_29_N_3_O_3_S: C, 62.50; H, 7.24; N,10.41; Found: C, 62.48; H, 7.22; N, 10.40.

Compound (**5s**): *(1R,2S,5R)-2-isopropyl-5-methylcyclohexyl-2-((5-(2-aminophenyl)-4-methyl-4H-1,2,4-triazol-3-yl)thio) acetate*. Pale yellow liquid. Yield: 59.6%; IR (KBr, cm^−1^): 3454, 3351 (m, N-H), 2954, 2928, 2869 (s, C-H), 1729 (s, C=O), 1469 (s, C=N), 707 (w, C-S-C); ^1^H NMR (500 MHz, CDCl_3_) δ 7.22 (td, *J* = 7.9, 1.5 Hz, 1H, C_20_-H), 7.17 (dd, *J* = 7.7, 1.5 Hz, 1H, C_18_-H), 6.82–6.76 (m, 2H, C_17_-H, C_19_-H), 4.84 (s, 2H, NH_2_-H), 4.72 (td, *J* = 10.9, 4.4 Hz, 1H, C_1_-H), 4.09 (s, 2H, C_12_-H), 3.61 (s, 3H, C_21_-H), 1.99 (dtd, *J* = 12.1, 4.1, 1.7 Hz, 1H, C_6_-H_a_), 1.85 (pd, *J* = 7.0, 2.8 Hz, 1H, C_7_-H), 1.68 (q, *J* = 2.7, 2.1 Hz, 1H, C_4_-H_a_), 1.65 (q, *J* = 2.8 Hz, 1H, C_3_-H_a_), 1.46 (dddd, *J* = 12.1, 8.9, 6.2, 3.2 Hz, 1H, C_5_-H), 1.39 (ddt, *J* = 14.2, 10.9, 3.1 Hz, 1H, C_2_-H), 1.04 (ddd, *J* = 15.7, 9.7, 3.1 Hz, 1H, C_3_-H_b_), 1.01–0.95 (m, 1H, C_6_-H_b_), 0.88 (t, *J* = 6.5 Hz, 6H, C_10_-H, C_8_-H or C_9_-H), 0.83 (dd, *J* = 12.3, 3.3 Hz, 1H, C_4_-H_b_), 0.73 (d, *J* = 6.9 Hz, 3H, C_8_-H or C_9_-H); ^13^C NMR (125 MHz, CDCl_3_) δ 168.01, 154.57, 150.25, 146.68, 131.11, 129.14, 117.40, 116.59, 110.43, 76.39, 46.93, 40.62, 35.68, 34.11, 32.04, 31.41, 26.19, 23.37, 21.96, 20.74, 16.28; ESI-MS *m*/*z*: 403.23 [M + H]^+^. Anal. Calcd. for C_21_H_30_N_4_O_2_S: C, 62.66; H, 7.51; N, 13.92; Found: C, 62.65; H, 7.50; N, 13.90.

Compound (**5t**): *(1R,2S,5R)-2-isopropyl-5-methylcyclohexyl-2-((5-(2-aminophenyl)-4-methyl-4H-1,2,4-triazol-3-yl)thio)acetate*. Pale yellow liquid. Yield: 52.7%; IR (KBr, cm^−1^): 3452, 3348 (m, N-H), 2954, 2927, 2868 (s, C-H), 1728 (s, C=O), 1469 (s, C=N), 702 (w, C-S-C); ^1^H NMR (500 MHz, CDCl_3_) δ 7.40 (d, *J* = 7.9 Hz, 2H, C_16_-H, C_20_-H), 6.74 (d, *J* = 8.1 Hz, 2H, C_17_-H, C_19_-H), 4.70 (td, *J* = 10.9, 4.4 Hz, 1H, C_1_-H), 4.04 (s, 2H, C_12_-H), 3.65–3.59 (m, 3H, C_21_-H), 2.95 (s, 2H, NH_2_-H), 2.00–1.94 (m, 1H, C_6_-H_a_), 1.81 (td, *J* = 7.0, 2.7 Hz, 1H, C_7_-H), 1.67 (d, *J* = 3.3 Hz, 1H, C_4_-H_a_), 1.64 (q, *J* = 2.9 Hz, 1H, C_3_-H_a_), 1.45 (dddt, *J* = 12.0, 8.8, 6.2, 3.4 Hz, 1H, C_5_-H), 1.37 (ddt, *J* = 14.4, 10.9, 2.8 Hz, 1H, C_2_-H), 1.08–1.00 (m, 1H, C_3_-H_b_), 1.00–0.93 (m, 1H, C_6_-H_b_), 0.90–0.85 (m, 6H, C_10_-H, C_8_-H or C_9_-H), 0.85–0.79 (m, 1H, C_4_-H_b_), 0.72 (d, *J* = 7.0 Hz, 3H, C_8_-H or C_9_-H); ^13^C NMR (125 MHz, CDCl_3_) δ 168.17, 156.46, 149.75, 148.34, 129.88, 116.36, 114.84, 76.31, 46.91, 40.62, 36.12, 34.11, 31.85, 31.40, 26.16, 23.35, 21.97, 20.75, 16.27; ESI-MS *m*/*z*: 403.23 [M + H]^+^. Anal. Calcd. for C_21_H_30_N_4_O_2_S: C, 62.66; H, 7.51; N,13.92; Found: C, 62.64; H, 7.49; N, 13.91.

Compound (**5u**): *(1R,2S,5R)-2-isopropyl-5-methylcyclohexyl-2-((5-(4-(tert-butyl)phenyl)-4-methyl-4H-1,2,4-triazol-3-yl)thio)acetate.* White solid. Yield: 66.1%, m.p. 100.3–102.6 °C; IR (KBr, cm^−1^): 2955, 2926, 2871 (s, C-H), 1743(s, C=O), 1468 (s, C=N), 704 (w, C-S-C); ^1^H NMR (500 MHz, CDCl_3_) δ 7.58–7.54 (m, 2H, C_16_-H, C_20_-H), 7.53–7.49 (m, 2H, C_17_-H, C_19_-H), 4.71 (td, *J* = 10.9, 4.4 Hz, 1H, C_1_-H), 4.08 (s, 2H, C_12_-H), 3.66 (s, 3H, C_21_-H), 1.98 (dtd, *J* = 12.0, 4.0, 1.7 Hz, 1H, C_6_-H_a_), 1.84 (dd, *J* = 7.0, 2.8 Hz, 1H, C_7_-H), 1.68 (q, *J* = 2.6, 1.8 Hz, 1H, C_4_-H_a_), 1.65 (q, *J* = 2.8 Hz, 1H, C_3_-H_a_), 1.46 (tdd, *J* = 8.7, 5.9, 3.1 Hz, 1H, C_5_-H), 1.43–1.36 (m, 1H, C_2_-H), 1.35 (s, 9H, C_23_-H, C_24_-H, C_25_-H), 1.08–1.01 (m, 1H, C_3_-H_b_), 1.01–0.95 (m, 1H, C_6_-H_b_), 0.88 (t, *J* = 7.1 Hz, 6H, C_10_-H, C_8_-H or C_9_-H), 0.86 (s, 1H, C_4_-H_b_), 0.73 (d, *J* = 7.0 Hz, 3H, C_8_-H or C_9_-H); ^13^C NMR (125 MHz, CDCl_3_) δ 168.23, 156.28, 153.58, 150.41, 128.44, 126.01, 124.14, 76.46, 47.05, 40.76, 36.15, 35.02, 34.24, 31.97, 31.54, 31.32, 26.30, 23.49, 22.09, 20.88, 16.40; ESI-MS *m*/*z*: 444.28 [M + H]^+^. Anal. Calcd. for C_25_H_37_N_3_O_2_S: C, 67.68; H, 8.41; N, 9.47; Found: C, 67.66; H, 8.40; N, 9.46.

Compound (**5v**): *(1R,2S,5R)-2-isopropyl-5-methylcyclohexyl-2-((5-(3,4-dimethoxyphenyl)-4-methyl-4H-1,2,4-triazol-3-yl)thio)acetate.* Pale yellow liquid. Yield: 50.9%; IR (KBr, cm^−1^): 2955, 2926, 2869 (s, C-H), 1731 (s, C=O), 1422 (s, C=N), 686 (w, C-S-C); ^1^H NMR (500 MHz, CDCl_3_) δ 7.23 (d, J = 2.0 Hz, 1H, C_20_-H), 7.10 (dd, *J* = 8.3, 2.0 Hz, 1H, C_16_-H), 6.95 (d, *J* = 8.3 Hz, 1H, C_19_-H), 4.70 (td, *J* = 10.9, 4.4 Hz, 1H, C_1_-H), 4.06 (s, 2H, C_12_-H), 3.92 (d, *J* = 6.4 Hz, 6H, C_22_-H, C_23_-H), 3.66 (s, 3H, C_21_-H), 1.97 (dd, *J* = 11.9, 4.5 Hz, 1H, C_6_-H_a_), 1.81 (ddd, *J* = 14.0, 7.0, 2.8 Hz, 1H, C_7_-H), 1.69–1.66 (m, 1H, C_4_-H_a_), 1.64 (q, *J* = 2.8 Hz, 1H, C_3_-H_a_), 1.45 (dddd, *J* = 12.1, 8.9, 6.9, 3.1 Hz, 1H, C_5_-H), 1.37 (ddt, *J* = 14.2, 10.9, 3.1 Hz, 1H, C_2_-H), 1.08–1.00 (m, 1H, C_3_-H_b_), 1.00–0.94 (m, 1H, C_6_-H_b_), 0.87 (t, *J* = 7.0 Hz, 6H, C_10_-H, C_8_-H or C_9_-H), 0.85–0.80 (m, 1H, C_4_-H_b_), 0.72 (d, J = 7.0 Hz, 3H, C_8_-H or C_9_-H); ^13^C NMR (125 MHz, CDCl_3_) δ 168.11, 156.00, 150.62, 150.28, 149.34, 120.98, 119.48, 112.01, 110.99, 76.35, 56.09, 56.01, 46.93, 40.64, 36.08, 34.11, 31.93, 31.41, 26.17, 23.35, 21.97, 20.75, 16.28; ESI-MS *m*/*z*: 448.25 [M + H]^+^. Anal. Calcd. for C_23_H_33_N_3_O_4_S: C, 61.72; H, 7.43; N, 9.39; Found: C, 61.70; H, 7.42; N, 9.36.

Compound (**5w**): *(1R,2S,5R)-2-isopropyl-5-methylcyclohexyl-2-((5-(3,5-dimethoxyphenyl)-4-methyl-4H-1,2,4-triazol-3-yl)thio)acetate.* Pale yellow liquid. Yield: 62.7%; IR (KBr, cm^−1^): 2954, 2926, 2869 (s, C-H), 1731 (s, C=O), 1474 (s, C=N), 713 (w, C-S-C); ^1^H NMR (500 MHz, CDCl_3_) 6.75 (d, *J* = 2.3 Hz, 2H, C_16_-H, C_20_-H), 6.57 (t, *J* = 2.3 Hz, 1H, C_18_-H), 4.71 (td, *J* = 10.9, 4.4 Hz, 1H, C_1_-H), 4.07 (d, *J* = 1.3 Hz, 2H, C_12_-H), 3.82 (s, 6H, C_22_-H, C_23_-H), 3.66 (s, 3H, C_21_-H), 1.98 (dtd, *J* = 12.0, 4.0, 1.7 Hz, 1H, C_6_-H_a_), 1.82 (ddd, *J* = 11.2, 7.0, 3.5 Hz, 1H, C_7_-H), 1.67 (q, *J* = 2.6, 1.8 Hz, 1H, C_4_-H_a_), 1.66–1.62 (m, 1H, C_3_-H_a_), 1.46 (dddt, *J* = 12.1, 8.9, 6.2, 3.4 Hz, 1H, C_5_-H), 1.41–1.34 (m, 1H, C_2_-H), 1.08–1.01 (m, 1H, C_3_-H_b_), 1.00–0.95 (m, 1H, C_6_-H_b_), 0.90–0.86 (m, 6H, C_10_-H, C_8_-H or C_9_-H), 0.85–0.81 (m, 1H, C_4_-H_b_), 0.72 (d, *J* = 7.0 Hz, 3H, C_8_-H or C_9_-H); ^13^C NMR (125 MHz, CDCl_3_) δ 168.16, 161.15, 156.06, 150.76, 128.66, 106.82, 102.30, 76.49, 55.70, 47.04, 40.75, 36.10, 34.23, 32.06, 31.52, 26.29, 23.47, 22.08, 20.87, 16.39; ESI-MS *m*/*z*: 448.24 [M + H]^+^. Anal. Calcd. for C_23_H_33_N_3_O_4_S: C, 61.72; H, 7.43; N, 9.39; Found: C, 61.71; H, 7.40; N, 9.37.

Compound (**5x**): *(1R,2S,5R)-2-isopropyl-5-methylcyclohexyl-2-((5-(furan-2-yl)-4-methyl-4H-1,2,4-triazol-3-yl)thio)acetate.* White solid. Yield: 51.3%, m.p. 136.2–137.7 °C; IR (KBr, cm^−1^): 2961, 2935, 2866 (s, C-H), 1734 (s, C=O), 1474 (s, C=N), 707 (w, C-S-C); ^1^H NMR (500 MHz, CDCl_3_) δ 7.58 (t, *J* = 1.2 Hz, 1H, C_18_-H), 7.05 (d, *J* = 3.5 Hz, 1H, C_16_-H), 6.56 (dd, *J* = 3.5, 1.8 Hz, 1H, C_17_-H), 4.69 (td, *J* = 10.9, 4.4 Hz, 1H, C_1_-H), 4.03 (d, *J* = 2.3 Hz, 2H, C_12_-H), 3.83 (s, 3H, C_19_-H), 1.95 (dd, *J* = 10.8, 6.0 Hz, 1H, C_6_-H_a_), 1.78 (td, *J* = 6.9, 2.8 Hz, 1H, C_7_-H), 1.69–1.66 (m, 1H, C_4_-H_a_), 1.64 (q, *J* = 2.9 Hz, 1H, C_3_-H_a_), 1.44 (dddd, *J* = 13.0, 9.9, 6.6, 3.6 Hz, 1H, C_5_-H), 1.40–1.32 (m, 1H, C_2_-H), 1.07–0.99 (m, 1H, C_3_-H_b_), 0.99–0.93 (m, 1H, C_6_-H_b_), 0.86 (dd, *J* = 11.1, 6.8 Hz, 6H, C_10_-H, C_8_-H or C_9_-H), 0.84–0.79 (m, 1H, C_4_-H_b_), 0.71 (d, *J* = 6.9 Hz, 3H, C_8_-H or C_9_-H); ^13^C NMR (125 MHz, CDCl_3_) δ 168.10, 150.36, 148.42, 144.06, 142.43, 111.97, 111.91, 76.48, 47.02, 40.72, 36.35, 34.22, 32.08, 31.52, 26.26, 23.45, 22.06, 20.84, 16.35; ESI-MS *m*/*z*: 378.19 [M + H]^+^. Anal. Calcd. for C_19_H_27_N_3_O_3_S: C, 60.45; H, 7.21; N, 11.13; Found: C, 60.43; H, 7.20; N, 11.12.

Compound (**5y**): *(1R,2S,5R)-2-isopropyl-5-methylcyclohexyl-2-((4-methyl-5-(thiophen-2-yl)-4H-1,2,4-triazol-3-yl)thio)acetate.* White solid. Yield: 62.5%, m.p. 127.2–127.9 °C; IR (KBr, cm^−1^): 2952, 2934, 2915, 2864 (s, C-H), 1736 (s, C=O), 1474 (s, C=N), 707 (w, C-S-C); ^1^H NMR (500 MHz, CDCl_3_) δ 7.49 (dd, *J* = 5.1, 1.1 Hz, 1H, C_18_-H), 7.46 (dd, *J* = 3.7, 1.1 Hz, 1H, C_16_-H), 7.16 (dd, *J* = 5.1, 3.7 Hz, 1H, C_17_-H), 4.69 (td, *J* = 10.9, 4.4 Hz, 1H, C_1_-H), 4.04 (d, *J* = 2.0 Hz, 2H, C_12_-H), 3.76 (s, 3H, C_19_-H), 1.96 (ddq, *J* = 9.7, 3.9, 2.0 Hz, 1H, C_6_-H_a_), 1.79 (qd, *J* = 7.0, 2.8 Hz, 1H, C_7_-H), 1.67 (q, *J* = 2.6, 1.8 Hz, 1H, C_4_-H_a_), 1.66–1.62 (m, 1H, C_3_-H_a_), 1.45 (tdt, *J* = 12.2, 6.6, 3.3 Hz, 1H, C_5_-H), 1.39–1.34 (m, 1H, C_2_-H), 1.07–0.99 (m, 1H, C_3_-H_b_), 0.99–0.93 (m, 1H, C_6_-H_b_), 0.87 (dd, *J* = 9.0, 6.7 Hz, 6H, C_10_-H, C_8_-H or C_9_-H), 0.85–0.79 (m, 1H, C_4_-H_b_), 0.71 (d, *J* = 6.9 Hz, 3H, C_8_-H or C_9_-H); ^13^C NMR (125 MHz, CDCl_3_) δ 168.13, 151.15, 150.66, 128.49, 128.09, 127.90, 127.86, 76.50, 47.03, 40.73, 36.34, 34.21, 32.05, 31.52, 26.27, 23.45, 22.06, 20.85, 16.37; ESI-MS *m*/*z*: 394.16 [M + H]^+^. Anal. Calcd. for C_19_H_27_N_3_O_2_S_2_: C, 57.99; H, 6.92; N,10.68; Found: C, 57.98; H, 6.90; N, 10.66.

Compound (**5z**): *(1R,2S,5R)-2-isopropyl-5-methylcyclohexyl-2-((4-methyl-5-(pyridin-3-yl)-4H-1,2,4-triazol-3-yl)thio)acetate*. Yellow solid. Yield: 61.5%, m.p. 156.1–158.0 °C; IR (KBr, cm^−1^): 2962, 2945, 2916, 2866 (s, C-H), 1733 (s, C=O), 1481 (s, C=N), 712 (w, C-S-C); ^1^H NMR (500 MHz, CDCl_3_) δ 8.88 (dt, *J* = 2.1, 1.1 Hz, 1H, C_19_-H), 8.73 (dd, *J* = 4.1, 2.4 Hz, 1H, C_18_-H), 8.04–7.99 (m, 1H, C_16_-H), 7.45 (ddt, *J* = 7.9, 4.9, 1.1 Hz, 1H, C_17_-H), 4.71 (td, *J* = 11.0, 1.3 Hz, 1H, C_1_-H), 4.09 (s, 2H, C_12_-H), 3.70 (s, 3H, C_20_-H), 1.97 (dt, *J* = 12.2, 4.0 Hz, 1H, C_6_-H_a_), 1.84–1.78 (m, 1H, C_7_-H), 1.67 (d, *J* = 3.2 Hz, 1H, C_4_-H_a_), 1.65 (t, *J* = 2.9 Hz, 1H, C_3_-H_a_), 1.45 (dt, *J* = 15.8, 5.3 Hz, 1H, C_5_-H), 1.41–1.34 (m, 1H, C_2_-H), 1.03 (ddd, *J* = 13.5, 10.4, 3.3 Hz, 1H, C_3_-H_b_), 1.00–0.95 (m, 1H, C_6_-H_b_), 0.87 (t, *J* = 7.0 Hz, 6H, C_10_-H, C_8_-H or C_9_-H), 0.85–0.80 (m, 1H, C_4_-H_b_), 0.72 (d, *J* = 7.0 Hz, 3H, C_8_-H or C_9_-H); ^13^C NMR (125 MHz, CDCl_3_) δ 168.05, 153.54, 151.60, 151.24, 148.96, 136.26, 123.89, 123.59, 76.60, 47.06, 40.76, 36.05, 34.21, 32.06, 31.52, 26.31, 23.47, 22.08, 20.86, 16.40; ESI-MS *m*/*z*: 389.19 [M + H]^+^. Anal. Calcd. for C_20_H_28_N_4_O_2_S: C, 61.83; H, 7.26; N,14.42; Found: C, 61.82; H, 7.25; N, 14.40.

### 3.4. Antifungal Activity Test

The primary biological activity of the menthol-based 1,2,4-triazole-thioether compounds was evaluated by agar dilution method against the tested fungi. The tested compound was dissolved in acetone. Sorporl-144 (200 mg/L), an emulsifier, was added to dilute the solution of each sample to 500 mg/L. Then, 50 µg/mL of the test compounds were obtained by mixing 1 mL of drug solution with 9 mL of PSA culture medium. A bacterium tray of 5-mm diameter cut along the external edge of the mycelium was transferred to the flat surface containing the tested compound and put in equilateral triangular style in triplicate. Later, the culture plates were cultivated at 24.0 ± 1.0 °C and the extended diameters of the circles of mycelium were measured after 48 h. Compared with the blank control, the relative inhibition percentage was calculated. All the experiments were performed in three replicates, and chlorothalonil (a commercial fungicide) was used as positive control. Activity grading indicators: Grade A: ≥90%; Grade B: 70~90%; Grade C: 50~70%; Grade D: <50%.

### 3.5. 3D-QSAR Analysis

The 3D-QSAR model for finding out the correlation between the structure and activity of the target compounds was built using the CoMFA method of Sybyl-X 2.1.1 software. According to the literature [41], the antifungal activities of the title compounds against *P. piricola* were transferred to ED values through the formula:
ED = lg {I/[(100 − I) × MW]}(1)
where I was the inhibitory percentage at 50 µg/mL and MW was the molecular weight of the target compounds. Complete conformational optimization of each structure was performed using a conjugate gradient procedure based on the Tripos force field with termination convergence energy of 0.005 kcal/(mol*Å) and Gasteiger-Hückel charges. Compound 5b (R = *o*-CH_3_ Ph) was selected as the template molecule, in which the atoms marked with asterisks constituted the common superimposed skeleton (Figure 4), twenty-three test compounds as training set were superimposed (Figure 5). CoMFA descriptors as independent variables and ED values as dependent variables. The cross-validation using the leave-one-out method was performed to obtain the cross-validated *q*^2^ and the optimal number *N* of components. Then, a non-cross-validation analysis under the optimal number of components was performed. The modeling capability was indicated by the correlation coefficient squared *r*^2^, and the prediction capability was judged by the *r*^2^ and *q*^2^. 

### 3.6. Molecular Docking Analysis

Molecular docking study was performed on Sybyl-X 2.1.1 program, using cytochrome P450 14α-sterol demethylase (CYP51, PDB ID 1EA1) as the receptor. The 3D structures of small-molecule ligands were obtained and minimized by the reported method [42]. Surflex-Dock was employed for the docking process, and the binding site of its original ligand was selected as the docking site. The docking result was also observed in the 2D diagrams drawn by LigPlot+ v2.1 (EMBL-European Bioinformatics Institute, Cambridge, United Kingdom).

## 4. Conclusions

In search of natural product-based antifungal agents, 26 novel menthol-based 1,2,4-triazole-thioether compounds were synthesized. Their structures were confirmed by FT-IR, NMR, ESI-MS, and elemental analysis. The in vitro antifungal activity of the target compounds **5a**–**5z** was preliminarily evaluated against eight plant pathogens at the concentration of 50 µg/mL. The bioassay results indicated that some of the target compounds showed good inhibitory activity against the tested fungi, especially against *P. piricola*. Compounds **5b** (R = *o*-CH_3_ Ph), **5i** (R = *o*-Cl Ph), **5v** (R = *m*,*p*-OCH_3_ Ph) and **5x** (R = *α*-furyl) had inhibition rates of 93.3%, 79.4%, and 79.4%, respectively, against *P. piricola*, much better than that of the positive control chlorothalonil. Compounds **5v** (R = *m*,*p*-OCH_3_ Ph) and **5g** (R = *o*-Cl Ph) held inhibition rates of 82.4% and 86.5% against *C. arachidicola* and *G. zeae*, respectively, much better than that of the commercial fungicide chlorothalonil. Compound **5b** (R = *o*-CH_3_ Ph) displayed antifungal activity of 90.5% and 83.8%, respectively, against *C. orbicalare* and *F. oxysporum* f. sp. *Cucumerinu*. Compound **5m** (R = *o*-I Ph) had inhibition rates of 88.6%, 80.0%, and 88.0%, respectively, against *F. oxysporum* f. sp. *Cucumerinu*, *B. myadis* and *C. orbiculare*. Furthermore, compound **5b** (R=*o*-CH_3_ Ph) displayed the best and broad-spectrum antifungal activity against all the tested fungi. To design more effective antifungal molecules against *P. piricola*, 3D-QSAR analysis was carried out using CoMFA method. A reasonable and effective 3D-QSAR model (*r*^2^ = 0.991, *q*^2^ = 0.514) has been established. Molecular docking study revealed that there were H bonds and hydrophobic interactions between the target compounds and CYP51. Overall, compound **5b** (R = *o*-CH_3_ Ph) was a potential CYP51 inhibitor worthy of further study.

## Data Availability

Data are contained within the article or the Appendix A.

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
