# Peer review of "Synthesis, Antifungal Activity, 3D-QSAR, and Molecular Docking Study of Novel Menthol-Derived 1,2,4-Triazole-thioether Compounds"

_molecules, 2021, doi:10.3390/molecules26226948_

Round 1

Reviewer 1 Report

Some major concerns are needed to address.

  1. In the abstract, Line 17, Physalospora piricola should be Italic.
  2. The introduction is too general and should be improved. Better to focus on the actual fact of this research.
  3. Results and discussion: In the case of synthesis and characterization, the results of FT-IR, H-NMR, and C-NMR were discussed in order to prove the successful synthesis. Although, all these data had been shown in the supplementary materials but were not cited in the main manuscript. In this regard, how can a reader understand the necessity of these data? In addition, other data like ESI-MS and elemental analysis were also not included in the manuscript. These might produce some confusion among the readers.
  4. In the methods and materials, the author mentioned “50 mg/L of the test compounds were obtained by mixing 1 mL of drug solution with 9 mL of PSA culture medium”. But in the abstract and results section the author mentioned 50 µg/mL for antifungal analysis. I think the overall research is contradictory in this regard.
  5. All tested compound was dissolved in acetone. However, the author did not mention the antifungal effect of acetone.
  6. For molecular docking analysis, more details are needed regarding the methodology of the docking simulations. How many runs? How was the grid box size compared to the protein? Was the box placed at different positions? Where did the dockings run in multiple batches with random starting configuration? What was the criterion for the choice of the docked configuration? Was docking configuration clusters considered in determining the docking configuration?
  7. Molecular docking data and analysis is not comprehensive as a reader I am not able to obtain any conclusion or valuable information out of it. Please try to be concise and conclusive as much as possible (doi.org/10.3390/biology10080703).

Author Response

  1. In the abstract, Line 17, Physalospora piricola should be Italic.

Answer: Thank you for your reminding. The related content has been revised. (see line 17 in the abstract)

  1. The introduction is too general and should be improved. Better to focus on the actual fact of this research.

Answer: The main text of the “Introduction” section has been revised according to your comment. (see lines 34-37 in page 1)

  1. Results and discussion: In the case of synthesis and characterization, the results of FT-IR, H-NMR, and C-NMR were discussed in order to prove the successful synthesis. Although, all these data had been shown in the supplementary materials but were not cited in the main manuscript. In this regard, how can a reader understand the necessity of these data? In addition, other data like ESI-MS and elemental analysis were also not included in the manuscript. These might produce some confusion among the readers.

Answer: It has been added into the main manuscript. (see line 79 in page 2)

  1. In the methods and materials, the author mentioned “50 mg/L of the test compounds were obtained by mixing 1 mL of drug solution with 9 mL of PSA culture medium”. But in the abstract and results section the author mentioned 50 µg/mL for antifungal analysis. I think the overall research is contradictory in this regard.

Answer: It has been revised. (see line 592 in page 15)

  1. All tested compound was dissolved in acetone. However, the author did not mention the antifungal effect of acetone.

Answer: In the bioassay test, the relative inhibition percentages of the target compounds was calculated by comparison of the mycelium diameter of the treatment group with that of the blank control, which could eliminate the effect of acetone on the results. It has been supplementarily clarified in the related text. (see lines 597-599 in page 15)

  1. For molecular docking analysis, more details are needed regarding the methodology of the docking simulations. How many runs? How was the grid box size compared to the protein? Was the box placed at different positions? Where did the dockings run in multiple batches with random starting configuration? What was the criterion for the choice of the docked configuration? Was docking configuration clusters considered in determining the docking configuration?

Answer: Thank you for your advice. In our work, molecular docking study was carried out by using Sybyl-X 2.1.1 software, an automated docking program. The information about how to select the docking site has been added in the main text. (see lines 630-631 in page 16), and other details about the docking procedure were as the default settings of the program. For the choice of the docked configuration, this program would automatically choose and present the one with the highest total score.

  1. Molecular docking data and analysis is not comprehensive as a reader I am not able to obtain any conclusion or valuable information out of it. Please try to be concise and conclusive as much as possible (org/10.3390/biology10080703).

Answer: Thank you for this enlightening comment. For comparison, the binding modes of imibenconazole presented in the 2D pattern were supplemented into the Figure 3, and the corresponding discussion was also improved. (see lines 184-197 in pages 7-8; see lines 763-765 in page 19)

Reviewer 2 Report

The manuscript molecules-1454738 entitled “Synthesis, Antifungal Activity, 3D-QSAR and Molecular Docking Study of Novel Menthol-Derived 1,2,4-Triazole-Thioether Compounds” by W.-G. Duan, G.-S. Lin et al. reports the synthesis of 25 new compounds based on (-)-menthol linked to 1,2,4-triazole-thioether moiety. All the obtained compounds have been characterized by IR, NMR and mass analyzes and the studies show a good antifungal inhibitory activity on some selected compounds against eight pathogenic plants, although the 50 µg/ml concentration used is not low. No IC50 data is reported for these compounds and therefore their toxicity cannot be estimated. 3D-QSAR analysis 630 was carried out and useful information for further inhibition studies  were reported. The manuscript is well written and all data is well organized, but some inaccuracies need to be fixed.

My opinion on the manuscript is to be considered after a major revision on the points I indicate below.

  1. No information on the preparation of derivatives 3 and 4 is provided. Considering that they are known precursors, at least indicate the bibliographic reference of the procedure reported in the literature for each product inserted in the manuscript.
  2. Sulfur and fluorine when present are missing in all elemental analysis data, furthermore, the presence of incorrect data suggests that the experimental analyzes were not actually carried out. I suggest deleting any data that has not actually been experimentally analyzed.
  3. The mass data of some derivatives are wrong with respect to the spectrum reported in supporting information, for 5a, 5l and 5q.
  4. The mass spectra of 5b and 5c coincide, one of the two needs to be re-examined.
  5. The calculated elemental analysis data of 5l are wrong and consequently the experimental data should not coincide.
  6. The formula of 5o, 5p, 5q and 5z are wrong.
  7. The elemental analysis of 5q is incorrect.

Here some typos in the text:

Line 35: correct “madicine” with “medicine”.

Line 39: “Actually” should be removed.

Lines 52-53: “because of their various bioactive activities such as insecticidal” these characters appear to have different formatting.

Line 62: correct in “(-)-menthyl-2-chloroacetate”.

Lines 159-161: correct this sentence  “The blue block represented that the introduction of electron-donating groups was favor to improve activity, and the red block was 159 opposite.” with “The blue block represents that the introduction of electron donor groups is conducive to enhancing activity and the opposite for the red block.”

Line 210: the number “2” in bold.

Line 220: correct in “menthyl-2-chloroacetate”.

Line 345: correct formatting of “methylcyclohexyl” that results in subscript.

Author Response

  1. No information on the preparation of derivatives 3 and 4 is provided. Considering that they are known precursors, at least indicate the bibliographic reference of the procedure reported in the literature for each product inserted in the manuscript.

Answer: Thank you for your advice. The related contents have been revised. (see lines 215-217 in page 8; line 235 in page 8; lines 768-770 in page 19)

  1. Sulfur and fluorine when present are missing in all elemental analysis data, furthermore, the presence of incorrect data suggests that the experimental analyzes were not actually carried out. I suggest deleting any data that has not actually been experimentally analyzed.

Answer: In this work, all of the target compounds were newly synthesized. For those newly synthetic compounds, elemental analysis was indispensable for their structural confirmation. Generally, the elemental analysis data of the target compounds only present the percentage contents of C, H, and N elements.

  1. The mass data of some derivatives are wrong with respect to the spectrum reported in supporting information, for 5a, 5l and 5q.

Answer: The related contents have been revised. (see line 256 in page 9; line 400 in page 11; line 467, page 13)

  1. The mass spectra of 5b and 5c coincide, one of the two needs to be re-examined.

Answer: The related contents have been revised.(see the Supplementary Materials)

  1. The calculated elemental analysis data of 5l are wrong and consequently the experimental data should not coincide.

Answer: The related contents have been revised. (see lines 400-401 in page 11)

  1. The formula of 5o, 5p, 5q and 5z are wrong.

Answer: The related contents have been revised. (see lines 440 and 453 in page 12; line 468 in page 13; line 586 in page 15)

  1. The elemental analysis of 5q is incorrect.

Answer: The related content has been revised. (see line 468 in page 13)

  1. Line 35: correct “madicine” with “medicine”.

Answer: The related content has been revised. (see line 39 in page 1)

  1. Line 39: “Actually” should be removed.

Answer: Thank you for your reminding. “Actually” has been removed.

  1. Lines 52-53: “because of their various bioactive activities such as insecticidal” these characters appear to have different formatting.

Answer: The related contents have been revised. (see lines 54-56 in page 2)

  1. Line 62: correct in “(-)-menthyl-2-chloroacetate”.

Answer: The related content has been revised. (see line 66 in page 2)

  1. Lines 159-161: correct this sentence “The blue block represented that the introduction of electron-donating groups was favor to improve activity, and the red block was 159 opposite.” with “The blue block represents that the introduction of electron donor groups is conducive to enhancing activity and the opposite for the red block.”

Answer: The related expression has been revised. (see lines 162-164, page 6)

  1. Line 210: the number “2” in bold.

Answer: The related content has been revised. (see line 219 and 227, page 8)

  1. Line 220: correct in “menthyl-2-chloroacetate”.

Answer: The related content has been revised. (see line 219, page 8)

  1. Line 345: correct formatting of “methylcyclohexyl” that results in subscript.

Answer: The related content has been revised. (see line 351, page 11)

Reviewer 3 Report

The target compounds were tested against plant pathogens.

The title of the manuscript should contain this information.

Also this information should be outlined in the abstract section to outline that the titled compound was designed for treatment of pathogen attacking plant but not human to avoid misunderstanding and to clarify the rational of the study .

In the experimental, the authors stated that they used 50 mg/l of the target compound for inhibiting fungi. However they stated in the conclusion section that the used concentration of the tested compounds was 50 µg/ml. What is the correct concentration?

The authors stated that they used GC analysis for indicating the chirality of the synthesized compounds. What was the used method for determination of GC? The authors needs to write the enantiomeric excess (ee) ratio in experimental section.

Why P450 14α-sterol demethylase (CYP51, PDB ID 1EA1) as the receptor was selected for molecular docking?

Author Response

  1. The target compounds were tested against plant pathogens. The title of the manuscript should contain this information. Also, this information should be outlined in the abstract section to outline that the titled compound was designed for treatment of pathogen attacking plant but not human to avoid misunderstanding and to clarify the rational of the study.

Answer: Thank you for your suggestion. In fact, the word “fungi” refers to plant pathogens, so the word “antifungal” has the same meaning as anti-plant pathogens. Following are three papers published in Molecules by our research group:

  • Ming Chen, Wen-Gui Duan*, Gui-Shan Lin*, Zhong-Tian Fan, Xiu Wang. Synthesis, Antifungal Activity and 3D-QSAR Study of Novel Nopol-Derived 1,3,4-Thiadiazole-Thiourea Compounds. Molecules, 2021, 26(6): 1708-1723
  • Guo-Qiang Kang, Wen-Gui Duan*, Gui-Shan Lin*, You-Pei Yu, Xiao-Yu Wang, Sun-Zhong Lu. Synthesis of Bioactive Compounds from 3-Carene (II): Synthesis, Antifungal Activity and 3D-QSAR Study of (Z)- and (E)-3-Caren-5-One Oxime Sulfonates. Molecules, 2019, 24(3): 477 -490
  • Gui-Shan Lin,Wen-Gui Duan*,Lin-Xiao Yang,Min Huang,Fu-Hou Lei.Synthesis and Antifungal Activity of Novel Myrtenal-Based 4-Methyl-1,2,4-triazole-thioethers. Molecules, 2017, 22(2):193-202

  1. In the experimental, the authors stated that they used 50 mg/l of the target compound for inhibiting fungi. However, they stated in the conclusion section that the used concentration of the tested compounds was 50 µg/ml. What is the correct concentration?

Answer: Thank you for your reminding. It has been revised. (see line 592 in page 15)

  1. The authors stated that they used GC analysis for indicating the chirality of the synthesized compounds. What was the used method for determination of GC? The authors needs to write the enantiomeric excess (ee) ratio in experimental section.

Answer: In this work, L-(-)-menthol, with chemical name of (1R,2S,5R)-2-isopropyl-5-methylcyclohexanol, was used as starting material. GC analysis was only employed to determine the purity of the starting material L-(-)-menthol, not to determine the chirality of the synthesized compounds. Actually, in the synthesis of the target compounds, the configuration of the three chiral carbons in the starting material remains unchanged, so it is not necessary to determine the ee values of the target compounds.

  1. Why P450 14α-sterol demethylase (CYP51, PDB ID 1EA1) as the receptor was selected for molecular docking?

Answer: 1,2,4-Triazole served as an important pharmacophore in the structures of the target compounds. As described in the “introduction” section, (see line 54-56 in page 2) many derivatives, which were recognized as cytochrome P450 14α-sterol demethylase (CYP51) inhibitors in the FRAC classification of fungicides (http://www.frac.info/), were found to bear 1,2,4-triazole moiety. Therefore, it could be deduced that, cytochrome P450 14α-sterol demethylase (CYP51) was a representative potential target for these title compounds containing 1,2,4-triazole moiety.

Round 2

Reviewer 1 Report

The author has substantially improved the manuscript.

Reviewer 2 Report

The authors responded to my comments and reviewed the reported incorrect data. My opinion of the manuscript after review is to be accepted in present form for publication on Molecules.

Reviewer 3 Report

The authors had answered the comments. However they need to do the following:

They should write a sentence at the abstract to state the rational of the manuscript. The abstract section to outline that the titled compound was designed for treatment of pathogen attacking plant.

Also, at the introduction section, the author need to write about the method for treatment of the target fungi and treatment of plants from Physalospora piricola pathogen

Is chlorothalonil used for treatment Physalospora piricola?

Concerning GC analysis for the starting material, the authors should state in the experimental section about this information.

The manuscript can be accepted after minor revision